# Dynamically chiral phosphonic acid-type metallo-β-lactamase inhibitors
Kinga Virág Gulyás[1], Liping Zhou[2], Daniel Salamonsen[3], Andreas Prester [4], Kim Bartels[4], Robert Bosman[4], Paul Haffke[4], Jintian Li[2], Viola Tamási[5], Fritz Deufel [1], Johannes Thoma [6], Anna Andersson Rasmussen[7], Miklós Csala [5], Hanna-Kirsti Schroder Leiros [3], Zhijian Xu [2], Mikael Widersten [1], Holger Rohde [4], Eike C. Schulz [4,8,9], Weiliang Zhu [2] & Máté Erdélyi [1,10] ✉

Antibiotic resistance is a growing global health threat that risks the lives of millions. Among the resistance mechanisms, that mediated by metallo-β-lactamases is of particular concern as these bacterial enzymes dismantle most β-lactam antibiotics, which are our widest applied and cheapest to produce antibiotic agents. So far, no clinically applicable metallo-β-lactamase inhibitors are available. Aiming to adapt to structural variations, we introduce the inhibitor concept: dynamically chiral phosphonic acids. We demonstrate that they are straightforward to synthesize, penetrate bacterial membranes, inhibit the metallo-β-lactamase enzymes NDM-1, VIM-2 and GIM-1, and are non-toxic to human cells. Mimicking the transition state of β-lactam hydrolysis, they target the Zn ions of the metallo-β-lactamase active site. As a unique feature, both of their stereoisomers bind metallo-β-lactamases, which provides them unparalleled adaptability to the structural diversity of these enzymes, and may allow them to hamper bacteria's ability for resistance development.

Antibiotic resistance poses a severe threat to global public health, necessitating the development of novel therapeutic strategies[1–3]. Among the resistance mechanisms, the production of metallo-β-lactamases represents a particularly formidable challenge[4–6]. These bacterial enzymes confer resistance to most β-lactam antibiotics, including carbapenems, which are among our last-resort treatments for multidrug-resistant Gram-negative bacterial infections[7–9].

Metallo-β-lactamase inhibitors may be used as adjunctive therapies that restore the efficacy of existing β-lactam antibiotics, and could thereby offer a viable strategy to address one of the most pressing health issues of current times[10–12]. So far, there are no metallo-β-lactamase inhibitors on the market. This leaves very limited therapeutic options (including polymyxins, tigecycline and aminoglycosides) to treat β-lactam-resistant bacterial infections, which are suboptimal as they often confer serious toxic side effects including the constriction of bronchioles, anaphylaxis, and kidney and neurological disorders[9,13,14].

A number of compounds[15–18], including captopril derivatives, have been proposed as metallo-β-lactamase inhibitors. Most have IC$_{50}$s in the μM range, even if a few exceptions with nM potency have also been reported[15]. The two most successful candidates, xeruborbactam (in clinical trials; IC$_{50}$ 0.1 μM for VIM-2; 4.3 μM for NDM-1) and taniborbactam (finished clinical trials; IC$_{50}$ 0.04 μM for VIM-2; 0.1 μM for NDM-1) are both bicyclic boronates[11,19–21]. These coordinate to the Zn1 ion with their boronate hydroxy group and to the Zn2 ion of the active site with their carboxylate group, which are their key interactions for binding[19]. The effectiveness of bicyclic boronates relies on the orientation of their boron atom towards the nucleophilic hydroxide, which direction is influenced by the catalytic residues and the unique topology of the active site of each enzyme. Unfortunately, a single amino acid mutation is sufficient for bacteria to prevent binding of the inhibitors. This has been shown also for taniborbactam[19], against which resistance development has already been reported even before it could have reached the market.

Phosphonic acid-based transition state analogues have emerged as promising candidates for metallo-β-lactamase inhibition[22–26]. They bind tightly to and block the hydrolytic activity of metallo-β-lactamases, thereby restoring the susceptibility of resistant bacteria to β-lactam antibiotics[22,25,27].

[1]Department of Chemistry - BMC, Organic Chemistry and the Uppsala Antibiotic Center; Uppsala University, Uppsala, Sweden. [2]Drug Discovery and Design Center, Shanghai Institute of Materia Medica, Chinese Academy of Sciences, Shanghai, China. [3]Department of Chemistry, Faculty of Science and Technology, UiT The Arctic University of Norway, Tromsø, Norway. [4]University Medical Center Hamburg-Eppendorf (UKE), Hamburg, Germany. [5]Department of Molecular Biology, Semmelweis University, Budapest, Hungary. [6]Department of Chemistry & Molecular Biology, Center for Antibiotic Resistance Research, CARe, University of Gothenburg, Gothenburg, Sweden. [7]Lund Protein Production Platform, Lund, Sweden. [8]Max-Planck-Institute for Structure and Dynamics of Matter, Hamburg, Germany. [9]Institute for Nanostructure and Solid State Physics, Universität Hamburg, Hamburg, Germany. [10]Center of Excellence for the Chemical Mechanisms of Life, Uppsala University, Uppsala, Sweden. ✉e-mail: mate.erdelyi@kemi.uu.se

Despite their high potential, their clinical utility has so far been hindered by inefficient synthetic routes, poor aqueous solubility, and unsatisfactory inhibitory activity[26,28–30]. Overcoming these difficulties is expected to maximize their therapeutic potential.

Clearly, one of the key challenges in developing metallo-β-lactamase inhibitors lies in the structural diversity and flexibility of these enzymes[19]. Viable broad-range inhibitors need to have a capability to adapt to such structural variability, and preferably not depend on interactions to specific amino acid side chains of the protein. No strategy for the design of such adaptive inhibitors have yet been presented.

In light of the need for adaptability in inhibitor design, we are pleased to present an approach to inhibitor development: phosphonic acid-based stereodynamic inhibitors. These inhibitors are designed to adapt to structural variations of the metallo-β-lactamase active site, offering a promising solution for addressing the challenges posed by these enzymes.

Both interconverting stereoisomers bind the Zn ions of the enzyme active site without heavily relying on interactions to specific amino acid side chains of the protein. Their superior aqueous solubility is expected to provide improved pharmacokinetic properties, and the ease of their synthesis to enable convenient clinical development and large-scale production. Similar to most known metallo-β-lactamase inhibitory lead compounds, they show μM activity against the clinically most critical metallo-β-lactamase enzymes VIM-2, NDM-1[5,31], but also against GIM-1 that possesses the most different active site compared to that of all other enzymes of the B1 metallo-β-lactamase family[32,33]. GIM-1 has a more constrained and narrower binding pocket as compared to other metallo-β-lactamases, and its binding site is composed of aromatic residues instead of hydrophilic ones, yet also containing a Ser119 and a Glu121 (BBL numbering[34], Supplementary Table S1), which amino acid residues are not present at these positions in other metallo-β-lactamase enzymes. We anticipate that our stereodynamic design strategy will facilitate the development of broad-spectrum isoenzyme inhibitors and inhibitors that, due to their structural dynamics, are less susceptible to resistance development mediated by single point mutations.

## Results and discussion

### Stereodynamic inhibitor design provides adaptable inhibitors

Our design (Fig. 1a) capitalizes on the resemblance of the tetrahedral geometry of the phosphonic acid moiety to the sp³ intermediate of the hydrolysis of peptide bonds[27,30,35]. The core structure of our phosphonic acid-based transition state analogues mimics the hydrolysis product of β-lactam antibiotics; however, it is not further hydrolysable by metallo-β-lactamases, similar to the bicyclic boronate-type inhibitors. Once bound to the enzyme's active site, the phosphonic acid remains strongly coordinated to the Zn ions that are necessary for the enzymatic activity. This strong coordination inhibits the enzyme from cleaving β-lactam antibiotics. To enhance binding affinity, our inhibitors were designed to possess a benzene ring with a phenolic hydroxy group, analogous to that of the monobactam-type antibiotic nocardicin A, that are expected to engage in π-π stacking and hydrogen bonding interactions, respectively, within the catalytic site. Our design is further inspired by structural features of cyclic boronates, such as taniborbactam, the currently most successful metallo-β-lactamase inhibitor[11,36,37]. As finding the balance between aqueous solubility and lipophilicity is of key importance for bioavailability, our inhibitors feature a hydrophilic core with lipophilic and hydrophobic side chains, such as thiophenes and benzothiophenes with varying bulkiness, flexibility and polarity, attached via an amide bond allowing easy structural variability. Several amino acids of the active site of metallo-β-lactamases are hydrophobic, and accordingly, these hydrophobic moieties were expected to increase binding affinity[11,29,38]. Sulfurous heterocycles are not uncommon in β-lactam antibiotics and β-lactamase inhibitors, and are present for instance in vaborbactam, a non-β-lactam-type inhibitor[11].

Developing broad spectrum metallo-β-lactamase inhibitors is challenging, to a large extent due to differences in the loops encompassing the active site of the isoenzymes, and to the flexibility of some of the loops[8,37,39].

To address this dynamic template, our objective was to introduce an inhibitor concept, that provides unique adaptability. To this end, we incorporated a dynamically chiral[40] center next to the Zn binding motif. Drug candidates possessing a dynamically chiral stereocenter rapidly interconvert between multiple chiral configurations under physiological conditions, which allows adaptation of their stereochemistry for binding to a target. Stereodynamic compounds may convert to the stronger binding stereoisomer to bind to an enzyme's active site, or may allow binding with different, stereochemistry-dependent binding modes to the same enzymatic site. We anticipate that stereodynamics may facilitate inhibitor candidates' binding to a variety of metallo-β-lactamases, potentially even to mutated variants. We implemented dynamic chirality by incorporating an easily deprotonable stereocenter, which allows the interconversion of the enantiomeric forms via an attainable energetic barrier at physiological conditions. Introduction of a phenyl group adjacent to the α-proton of an amino acid has previously been shown to decrease the activation barrier of racemization by ~8 kcal mol$^{-1}$, accelerating racemization by a factor of 40,000[41]. A phosphonic acid moiety was incorporated to achieve strong binding to Zn ions, which are the key elements of the active site of metallo-β-lactamases. Comparably small hydrophobic moieties were chosen to promote binding of both interconverting enantiomers in the active site. The combination of these features was expected to provide structural adaptability that facilitates binding to the Zn-containing active site of a variety of metallo-β-lactamases[37], and that may help to prevent rapid resistance development upon single point mutations. We note that a drug candidate encompassing a mixture of stereoisomers that do not interconvert yet bind to the active site of an enzyme with different binding modes may provide a similar advantage.

### Straightforward synthesis provides a diverse set of phosphonic acids

We established a straightforward four steps synthetic route towards inhibitors 5a–m as shown in Fig. 1b, which was initiated by a Kabachnik-Fields reaction to form the protected (amino(phenyl)methyl)-phosphonic acid core structure. Subsequent deprotection of the amino group by hydrogenation, allows the formation of analogues, 4a–m, through amide coupling. As a virtually unlimited variety of carboxylic acids are commercially available, this step promotes easy structural diversification. Simultaneous deprotection of the phosphonic acid and the phenol functionalities provides the desired final products, in an overall 70% average yield. The final products are stereodynamic, thus mixtures of inseparable interconverting enantiomers (see details on the chiral separation in the **Supplementary Information, Section 5**).

### Broad spectrum metallo-β-lactamase inhibitory activity associated with no cytotoxicity

The inhibitory activity (IC$_{50}$, Fig. 2a, Supplementary Table S2 and K$_i$ in Supplementary Table S3) of 5a–m was evaluated against the bacterial metallo-β-lactamases VIM-2, GIM-1, and NDM-1 in an enzymatic assay. All compounds showed low μM (IC$_{50}$) inhibitory activity against at least one of these enzymes, and importantly, several of them inhibited more than one of the metallo-β-lactamases (Fig. 2a). No obvious correlation between structure of the amide attached moiety and the enzyme inhibitory activity was observed. However, we note that the most potent compounds, 5d, 5f, and 5g, all contain non-flexible and bulky moieties attached to the amide group (Fig. 1b). The orientation of this moiety of the less flexible inhibitors apparently plays a role, which is seen, for instance, when comparing the VIM-2 inhibitory activities of compounds 5d, 5f, 5g with those of 5c and 5e, respectively (Figs. 1b and 2a). A Lineweaver-Burk analysis[42] confirmed competitive inhibition (Fig. S40, for enzyme kinetics studies see **Section 7** in **Supplementary Information**). Importantly, none of the compounds showed significant cytotoxicity when evaluated against human HepG2 cells (Fig. 2a and Supplementary Table S7).

These enzyme inhibitory activities indicate that incorporating a phosphonic acid moiety into small molecule metallo-β-lactamase inhibitors

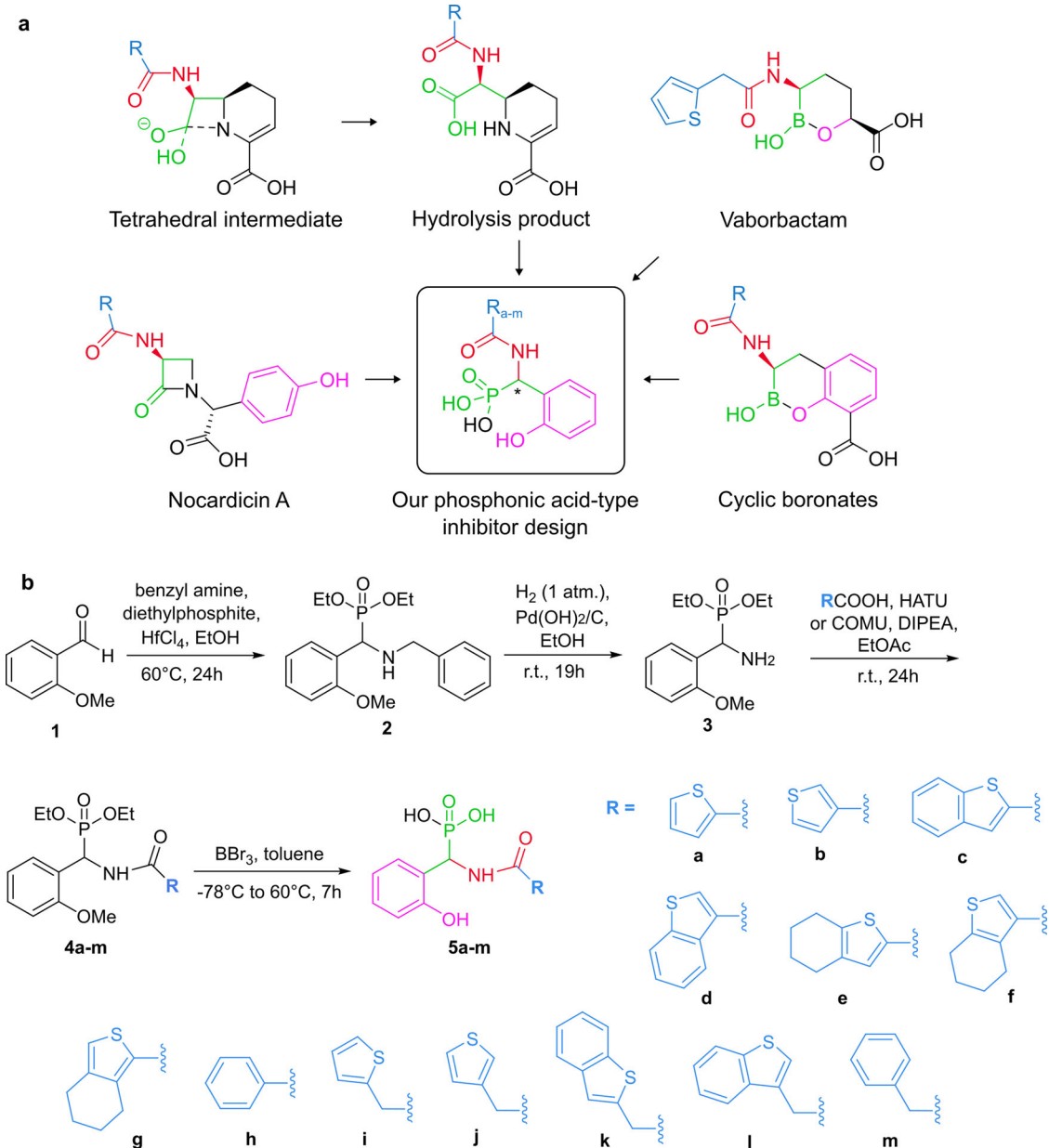

**Fig. 1 | Design and synthesis of the dynamically chiral phosphonic acid-based metallo-β-lactamase inhibitors. a** Our inhibitor design (circled) makes use of prior knowledge of the mechanism of β-lactam hydrolysis and of structural features of existing β-lactam antibiotics and β-lactamase inhibitors. **b** Synthetic route towards the stereodynamic phosphonic acid inhibitors 5a–m, which were designed to act as adaptive inhibitors of bacterial metallo-β-lactamase enzymes. This route provides straightforward variability of the moiety R connected via an amide bond. The acidity of the benzylic proton is responsible for the stereochemical dynamics. The phenyl (h) and benzyl (m) analogues are included to assess whether a sulfurous motif is beneficial for binding.

enhances their potency when compared to compounds that have no phosphonic acid group, or have a carboxylic acid group at a comparable position[23,25]. As phosphonates show affinity to metal ions, earlier approaches focused on combining a phosphonate group with other metal binding groups, such as thiols[27,43] and carboxylic acids[23,25,44] (Supplementary Table S8). In another approach, the phosphonic acid moiety was incorporated into β-lactam resembling structures, resulting in β-phospholactams[45] or their chemically more stable open analogue piperidinyl α-aminophosphonates[22]. The phosphonic acid-type inhibitors **5a–m** coordinate to the zinc ions, analogous to the previously reported phosphonates[22,23]. The structurally closest analogue piperidine-ring containing α-aminophosphonates showed IC$_{50}$ 4.1–328 μM and K$_d$ values 0.4–15 mM against VIM-2, and IC$_{50}$ 7.9–506 μM and K$_d$ of 0.5–3.1 mM against NDM-1[22]. β-Phospholactam, the analogue that best resembles a β-lactam so far, showed 53% inhibition of NDM-1 at 100 μM

concentration. The limited biological data available for this compound may be the consequence of demanding and low yielding synthesis, limited stability and low aqueous solubility. When connected to heteroaryl moiety, phosphonates show K$_i$ 0.3–30.3 μM against VIM-2, and 31.4–741.3 μM against NDM-1[23]. Heteroaryl 6-phosphonomethylpyridine-2-carboxylates encompassing several zink-binding motifs showed similar IC$_{50}$ values (0.464–1.90 μM) against VIM-2, and improved inhibition of NDM-1 (IC$_{50}$ 0.306–0.374 μM)[25]. The fully aliphatic N-(phosphonomethyl)-iminodiacetic acid, in which the phosphonic acid group is connected to two carboxylic acids through a nitrogen, was among the most potent compounds with IC$_{50}$ 0.68 μM against VIM-2 and 0.91 μM against NDM-1. It showed synergestic effect when combined with meropenem, and a high affinity to zinc ions (K$_d$ of 56 nM) due to zinc sequestration[25]. Furthermore, mercaptoethylphosphonates were presented as metallo-β-lactamase inhibitors with

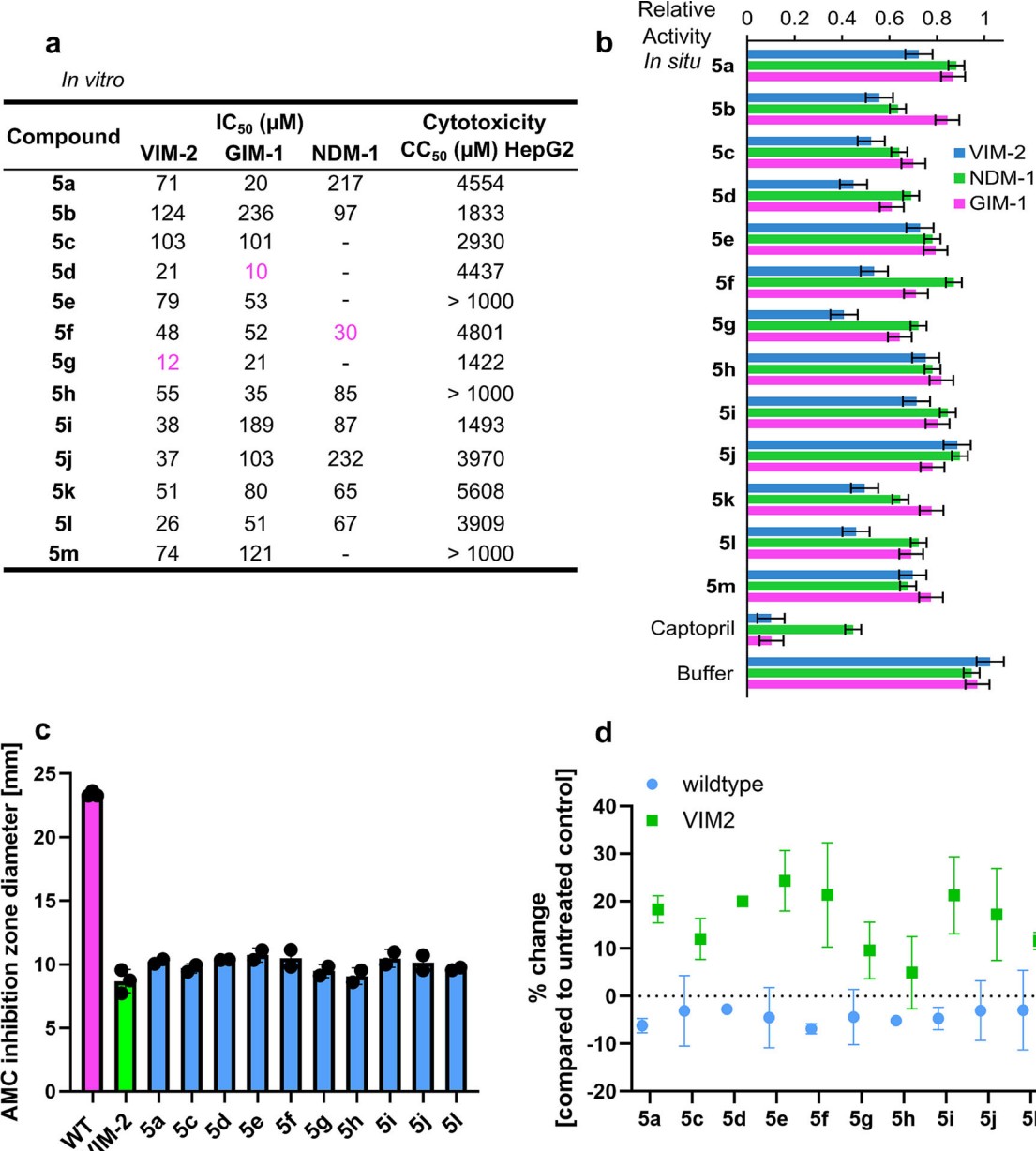

**Fig. 2 | Evaluation of VIM-2, GIM-1 and NDM-1 inhibitory activities of 5a–m.**
**a** Inhibitory activities (IC$_{50}$) against the purified metallo-β-lactamases VIM-2, GIM-1 and NDM-1, and cytotoxicities (CC$_{50}$) against HepG2 cells. Activities in the low µM range were observed for each of the three enzymes (in magenta), and several inhibitors showed activity against more than one of them. Some of the compounds showed absorbance interference with the substrate, which prevented the evaluation of their activity (no value given). Estimated inhibition constants (K$_i$) for VIM-2, GIM-1, and NDM-1 are given in Supplementary Table S3. **b** Normalized relative activity of metallo-β-lactamases encapsulated in outer membrane vesicles in absence (buffer) and presence of inhibitors. Error bars represent the standard deviation of four measurements. All compounds were used at a final concentration of 100 µM. **c**, **d** Effect of the inhibitors on ampicillin-clavulanic acid (AMC) growth inhibition zone diameters. AMC zone inhibition diameters were measured for *E. coli* TOP10 (wt, magenta column, **c**), *E. coli* TOP10xpCR4-Vim-2 (VIM-2, green column, **c**). Inhibitors 5a–l were added to AMC containing discs and inhibition zone diameters were recorded for *E. coli* TOP10 × pCR4-VIM-2 (blue columns, **c**). Columns represent the mean of two independent experiments. Several compounds show inhibition of VIM-2 activity (green squares) as compared to the control samples (blue circles, **d**). Error bars represent the standard deviation.

IC$_{50}$ 0.38–133 µM against VIM-2, 1.8–144 µM against NDM-1, and 0.18–>5000 µM against GIM-1[27]. X-ray crystallography indicated that only one of the stereoisomers of chiral mercaptophosphonates bind VIM-2, by coordination of its sulfur to the zinc ion of the enzyme whereas its phosphponate group interacts with the sidechain of Asn210 (Asn233, BBL[34] Supplementary Table S1)[27]. Overall, phosphonic acids and their close derivatives show high potential for use as metallo-β-lactamase inhibitors. Incorporation of a phosphonic acid moiety provides µM potency against metallo-β-

lactamases, which may be improved by further protein-binding moieties. Accordingly, phosphonates are a promising starting point for drug development.

## Enzyme inhibition inside outer membrane vesicles reveals membrane permeability

The ability of inhibitors **5a–m** to cross bacterial membranes was evaluated using outer membrane vesicles carrying VIM-2, NDM-1 or GIM-1 inside

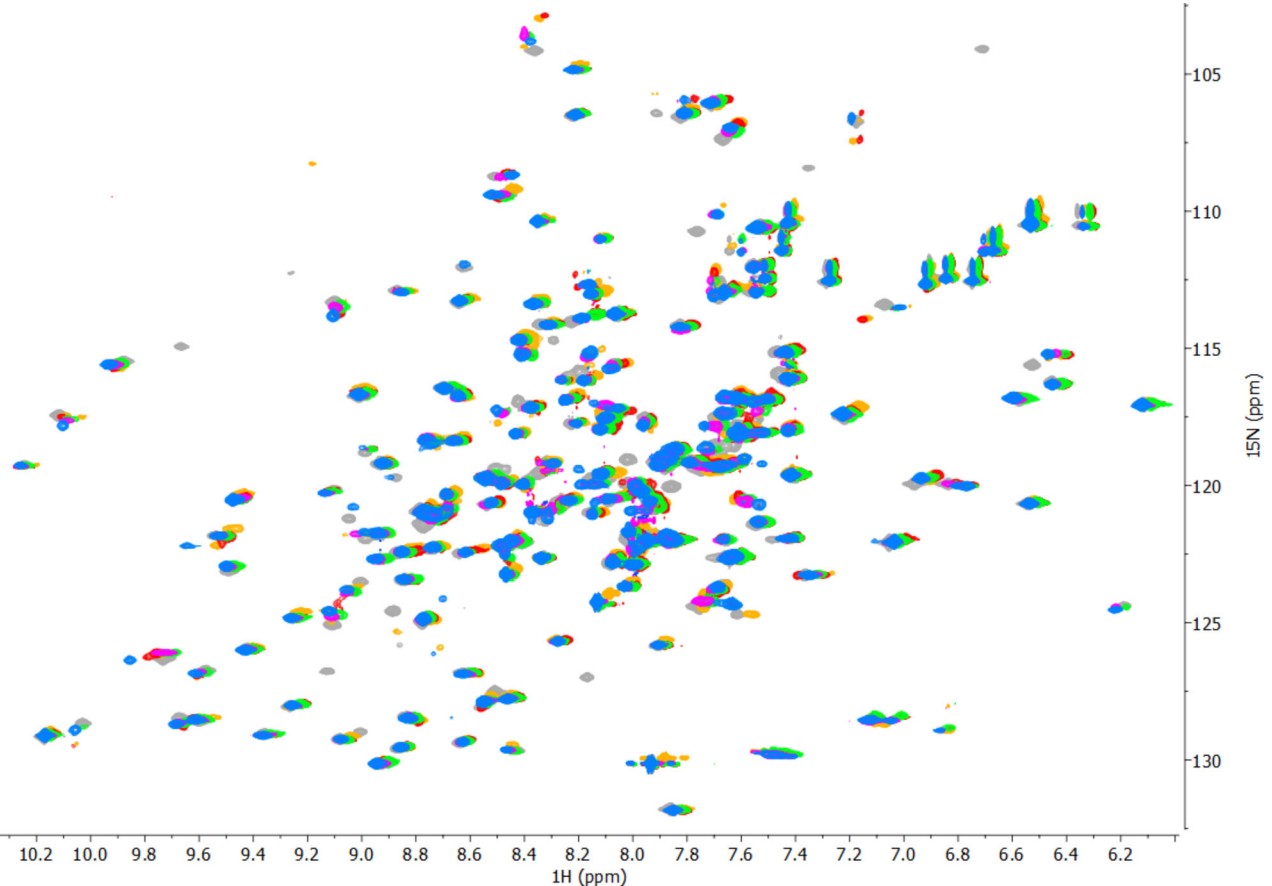

**Fig. 3 | Superimposition of the $^1$H, $^{15}$N-HSQC spectra (600 MHz, pH 7.0, 25 °C) of VIM-2 in the absence (gray) and in the presence of 10 molar equivalent of compound 5c (blue), 5d (red), 5g (yellow), 5i (green) and 5j (magenta).** The similarity of the chemical shift changes induced by the inhibitors suggest that the compounds bind in a comparable binding mode to VIM-2. No precipitation was observed throughout the titrations. Chemical shift perturbations were considered significant above the population mean plus at least one standard deviation[56,57].

their lumen[46]. Such outer membrane vesicles serve as an in situ model system for assessing transmembrane uptake, as their membrane maintains the protein as well as the lipid composition of the bacterial membrane they originate from[47]. The activity of metallo-β-lactamases encapsulated inside the vesicles was characterized by monitoring the hydrolysis of nitrocefin in the vesicles, and using buffer and L-captopril (IC$_{50}$ 4.4 μM)[48] as negative and positive references, respectively. As captopril's primary target is the angiotensin-converting enzyme, it is not used for metallo-β-lactamase inhibition clinically. However, due to its measurable activity against metallo-β-lactamases, it is a commonly applied reference compound in this context[48]. We observed a reduction in VIM-2, NDM-1 or GIM-1 activity in the vesicles in the presence of inhibitors **5a–m** (Fig. 2b), demonstrating that the studied compounds effectively cross the Gram-negative outer membrane. Upon permeabilization of the membrane with Triton X-100, the compounds showed comparable inhibitory activity on the free enzyme suggesting that the variation shown in Fig. 2b is not due to different permeability of **5a–m** (Supplementary Fig. S42).

**Phosphonate-type metallo-β-lactamase inhibitors influence bacteria**

To evaluate the enzyme inhibitory activity of ten inhibitors on living bacteria, we carried out disc diffusion experiments using a disc potentiation approach using antibiotic containing discs and Mueller–Hinton (MH) plates inoculated with *Escherichia coli* × pCR4-VIM-2. The addition of inhibitors resulted in an increase in inhibition zone diameters, ranging from 10 to 30% as compared to the control wild-type *E. coli* TOP10 without inhibitor (Fig. 2c, d). This indicates that the compounds show some VIM-2 inhibition in bacteria, corroborating the outcome of the enzyme inhibition

and membrane permeability assays. The extent of inhibition is however not directly clinically applicable. The computationally predicted ADME properties (**Section 11, Supplementary Information**) of **5a–m** suggest high bioavailability.

The lack of correlation between the inhibitory activities detected on isolated enzymes, the permeabilities detected on outer membrane vesicles, and activities on bacterial growth (Fig. 2) is not unexpected. The enzyme inhibition assay quantifies enzyme inhibition without further influencing factors, whereas membrane permeability measurements primarily indicate membrane permeability and do not quantify enzyme inhibition per se, whereas bacterial growth inhibition is influenced by a number of further factors, such as additional bacterial resistance mechanisms.

**Solution NMR further confirms enzyme–inhibitor binding**

Solution NMR spectroscopy provides understanding of enzyme-ligand interactions on an atomic level at near-physiological conditions, preserving the native state of enzymes[49]. To assess the binding event, we titrated $^{15}$N-labelled VIM-2[43] (for details about the labelled protein expression and purification, see **Section 15, Supplementary Information**) with up to 10 equivalents of inhibitors **5c, 5d, 5g, 5i,** and **5j**, following the binding induced chemical shift changes (**Section 12, Supplementary Information**). VIM-2 was chosen due to its high stability and to the overall stronger inhibitory activities observed towards this enzyme. Comparable $^1$H and $^{15}$N chemical shift changes throughout the titrations (Fig. 3) indicate that **5c, 5d, 5g, 5i,** and **5j** bind at the same binding cleft with similar binding mode. The data suggest that β-sheet 3, the hydrophobic loop 3 as well as loops-5, -10, and -12 and residues around the zinc coordination sites are involved in the binding event (the nomenclature follows ref. 38). The pattern and the magnitude of

the chemical shift changes are incompatible with Zn(II) depletion of the enzyme (over the 10 h experiment time)[22,29]. The obtained NMR data indicates reversible inhibition of the enzyme.

## X-ray diffraction confirms stereodynamic binding

For further analysis of the binding modes, we co-crystallized **5c**, **5d**, **5g**, and **5j** with VIM-2, and determined the structures of the complexes by X-ray diffraction with 1.34–1.92 Å resolution (Fig. 4, Supplementary Fig. S51 and Supplementary Table S28). In line with the NMR experiments, the X-ray data confirm that the inhibitors bind to the same binding cleft in analogous binding modes.

Common to all inhibitors, we observe the strongest interaction between the phosphonic acid moiety and the Zn ions, with a consistent O-Zn interatomic distance of ~1.9 Å. Depending on the stereochemistry of the bound inhibitor, hydrogen bonds to Asp118 or Asn210 (Asp120 and Asn233 according to BBL numbering, Supplementary Table S1[34]) may contribute to binding. Asp118 (Asp120, BBL[34]) may interact with the carbonyl oxygen or the phenolic hydroxy group, or the amide proton of some of the inhibitors. Asn210 (Asn233, BBL[34]) on the other hand can form a weak hydrogen bond either to the amide carbonyl, the phenolic hydroxy group or to one of the oxygens of the phosphonic acid of the inhibitor. The importance of Asn210 residue on metallo-β-lactamase inhibitor binding was previously also highlighted for homologous enzymes[50–52]. The number of hydrogen bonds to Asp118 or Asn210 (Asp120 and Asn233, BBL[34]) differs between the (R)- and (S)-enantiomers of an inhibitor. For inhibitor **5d**, a hydrogen bond is formed in the (S)-enantiomer between the phenyl group and the backbone amide of Asp118 (3.3 Å, Asp120, BBL[34]). In the (R)-enantiomer binding mode in subunit B, the phenyl- and the benzothiophene moieties have switched places, such that the inhibitor is rotated by ~180°. Thus, the carboxamido moiety is forming the hydrogen bond with the backbone amide of Asp118 (3.1 Å, Asp120, BBL[34]). The phenyl moiety forms an additional hydrogen bond with the side chain of Asn210 (3.1 Å, Asn233, BBL[34]). The (R)-enantiomer differs from the (S)-enantiomer in the additional hydrogen bond of the carboxamido moiety. However, the predominant interaction between VIM-2 and the inhibitors' variable hydrophobic moiety takes place via non-bonded contacts. Between the phenyl ring or the thiophene/benzothiophene moieties of the inhibitors, π-π stacking may be observed with the aromatic ring of Phe62, Tyr67 or Trp87 (Phe61, Tyr67 and Trp87, BBL[34]). Alternatively, a cation-π interaction of the benzothiophene moiety of the inhibitors **5c** and **5d** with Arg205 (Arg228, BBL[34]) of VIM-2 is found. Arginine is known to have a tendency to form strong cation-π interactions[53,54]. The guanidinium moieties of Arg205 (Arg228, BBL[34]) and the benzothiophene moieties of **5c** and **5d** are parallel to each other with a shortest distance of about 3.5–3.7 Å in the crystal structures. Based on the present results, we expect future structural modifications to allow stronger and more specific interactions to the amino acid side chains. The crystal structures of **5c** and **5d** unambiguously show that both, the (R)- and (S)-enantiomers bind to VIM-2 (Supplementary Table S30). Interestingly, the opposite enantiomer of the same inhibitor is bound rotated by ~180°.

We anticipate an analogous binding pattern for **5g** and **5j**, however, due to their highly dynamic nature, the electron density in these structures was conservatively better explained by only the (R)-enantiomer (**Section 13, Supplementary Information**). The sulfur of the thiophene/benzothiophene moieties does not provide a strong and specific interaction to the enzyme, as indicated by the comparable inhibitory activities and binding modes of **5h** and **5m** to the other sulfurous inhibitors.

NMR and X-ray diffraction unambiguously demonstrate that VIM-2 binds the stereodynamic inhibitors with a conserved primary interaction of their phosphonic acid to the Zn ions of the active site. Our results clearly underline the highly versatile binding modes of the inhibitors, thus confirming the primary design strategy that VIM-2 cannot only bind both enantiomers, but remarkably, is also tolerant to large-scale rotations in the active site as well as to structural variation of the inhibitors. Therefore, we hypothesize that these properties will aid preventing sudden resistance

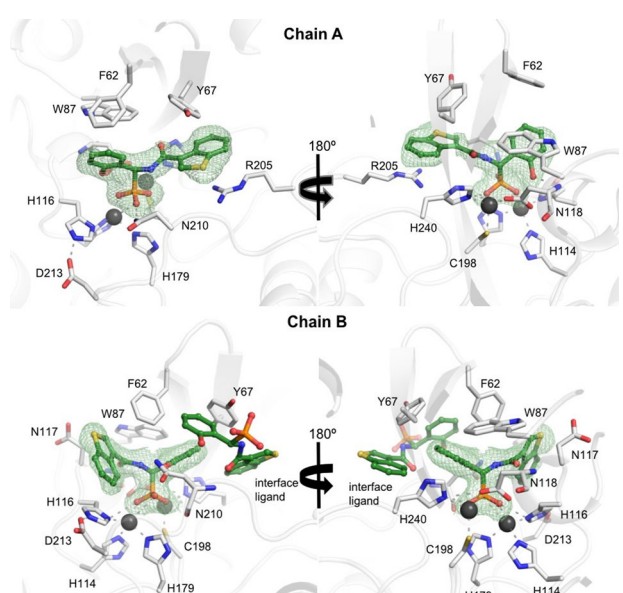

**Fig. 4 | VIM-2 crystal structure in complex with stereodynamic inhibitor 5d.** Chain A is shown bound to the (S)- whereas chain B to the (R)-stereoisomer of **5d**. Both the (R)- and (S)-enantiomers bind, but in different binding poses, rotated by ~180°. The protein is displayed as a white cartoon, interacting side chains and the inhibitor are shown as sticks and in ball-and-stick representation, respectively. A polder-omit electron density map is shown as green mesh at an rmsd of 3 Å, unambiguously demonstrating the absolute configuration of the inhibitors. Inhibitors **5c**, **5j**, and **5g** can be seen on Supplementary Fig. S51.

development as alternate binding modes could compensate for the structural changes caused by mutation. Capitalizing on this dynamic adaptability, will certainly be a fruitful route for further improvements of these inhibitors.

## Comparison of the binding mode to those of previous inhibitors and of hydrolyzed antibiotics

The structure of a variety of heteroaryl phosphonates in complex with VIM-2 (PDB IDs 6D15, 6D16, 6D17, 6D18, 6D19, 6D1A, 6D1B, 6D1C, 6D1D, 6D1E, 6D1F, 6D1G, 6D1H, 6D1I, 6D1J, 6D1K, 6DD0, 6DD1, 6OR3, and 6NY7) are available in the Protein Data Bank (RCSB PDB), however, enzyme inhibitory activity (MIC, IC$_{50}$) has so far been disclosed for [(5,7-dibromo-2-oxo-1,2-dihydroquinolin-4-yl)methyl]phosphonic acid (PDB 6O5T) only. The X-ray crystallographic structure of this inhibitor indicates that the phosphonic acid of the latter binds directly to Zn2 (2.1 Å) and via a hydroxide bridge (2.0 Å) to Zn1 of the VIM-2 binding site (Supplementary Fig. S54c). At low pH, however, the phosphonate binds to both Zn ions directly. Further hydrogen bonds of this inhibitor with the main chain amide of Asn210 (3.5 Å, Asn233, BBL[34]) and the side chain of Arg205 (3.5 Å, Arg228, BBL[34]) are observed, but no further strong interactions to the binding site are seen. In contrast to the inhibitors presented in this study, the π-stacking interactions are not as obvious from the crystallographic data of 6O5T. In only a few of the literature compounds have at most one additional hydrogen bond with the enzyme. Overall, the phosphonic acid moiety of the inhibitor in the 6O5T structure interacts with the Zn ions of the enzyme in a similar way as the phosphonic acid of **5c**, **5d**, **5j**, and **5g** whereas the stereodynamics of these inhibitors result in different binding modes as compared to the that of 5,7-dibromo-2-oxo-1,2-dihydroquinolin-4-yl)methyl]phosphonic acid (Supplementary Fig. S54).

6-(Phosphonomethyl)pyridine-2-carboxylic acid is a known inhibitor of the B1 metallo-β-lactamase IMP-1[44], with the co-crystal structure PDB 5HH4 indicating that its phosphonic acid interacts with the Zn1 ion of the binding site via a hydroxide ion (2.6 Å) whereas its pyridine nitrogen (2.7 Å) and carboxylate group (2.3 Å) bind the Zn2 ion (Supplementary Fig. S54d).

The binding mode of 6-(phosphonomethyl)pyridine-2-carboxylic acid to IMP-1 lacks direct phosponic acid – Zn contact and is thus fundamentally different from the binding poses of **5c**, **5d**, **5j**, and **5g** with VIM-2.

In the complex of VIM-2 with the hydrolysis product of biapenem (PDB 6Y6J), a carbapenem, the pyrrol (2.1 Å) and the carboxylate moiety (2.2 Å) of the hydrolyzed inhibitor closely interact with Zn2 of the binding site, whereas the carboxylate of the opened β-lactam ring interacts with Zn1 (1.9 Å) (Supplementary Fig. S55). Similar to the benzothiophene of **5c**, **5d**, **5j**, and **5g**, the heteroaromatic pyrazolotriazol moiety may engage in cation-π interaction. Further hydrogen bonds to the main chain amide of Asp118 (3.1 Å, Asp120, BBL[34]) and the sidechain of Asn210 (2.6 Å, Asn233, BBL[34]) resemble the binding pose of **5c**, **5d**, **5j**, and **5g**.

In serine β-lactamases, boronates react as Lewis acids with the active site serine and mimic the tetrahedral transition state of the catalytic mechanism[11]. In metallo-β-lactamases the boron atom of e.g., taniborbactam reacts with the active site hydroxide anion and thus adopts an sp[3] hybridization state. The boron hydroxy group interacts with the Asp118 and Asn210 (Asp120 and Asn233, BBL[34]) side chains as well as Zn1. Zn2, on the other hand, interacts with the carboxylate and the cyclic oxaborinanes oxygen atom of taniborbactam. Therefore, boronate-inhibitors such as taniborbactam mimic the tetrahedral intermediate in class B metallo-β-lactamases[55]. Akin to the boronate-inhibitors, our phosphonate β-lactamase inhibitors provide a phosphorus atom, which is also in a tetrahedral conformation. This is in contrast to approved boronate β-lactamase inhibitors: here the boronate moiety forms a cyclic tetrahedral moiety and therefore restricts potential orientations of the inhibitor molecule. This is clearly underlined by the taniborbactam structures, which adopt a similar orientation in both subunits (Supplementary Fig. S56c, d). In contrast, our dynamically chiral phosphonate inhibitors can rotate around the carbon phosphorus bond and adapt multiple orientations in the active site (Supplementary Fig. S56a, b). Moreover, due to its comparably increased length, taniborbactam interacts with Glu146 and Asp213 (Glu149 and Asp236, BBL[34]) and forms additional site-specific hydrogen bonds with the enzyme. The tetrahedral geometry is generally a short-lived state in the hydrolysis mechanism of β-lactam antibiotics. Inhibitors that stably adopt this geometry can therefore prevent the progress of hydrolysis and thus inhibit the enzyme. In summary, taniborbactam allows only one highly conserved binding mode while our phosphonic acid based inhibitors are more flexible due to their stereodynamic binding poses.

**Computational simulation confirms a conserved binding mode**

Performing full NMR and X-ray analyses for all combinations of inhibitors **5a–m** and the three studied enzymes would be unrealistic. We have therefore benchmarked a computational docking technique to the experimental data for the complexes of **5c**, **5d**, **5g**, and **5j** and VIM-2, and used the validated method to predict the structure of the complexes for which we have not obtained experimental data.

The predictive ability of the docking methodology was confirmed by comparison of the computationally generated and experimentally observed (X-ray) structures of the VIM-2 complexes of **5c**, **5d**, **5g**, and **5j** (Supplementary Fig. S62). The docking scores for VIM-2 were better than −8 kcal mol⁻¹, confirming that both stereoisomers bind VIM-2 with comparable affinity (Supplementary Table S31). Even the 180° rotated orientation of the stereoisomers of the inhibitors in the VIM-2 binding pocket was well-reproduced computationally, apart from **5g** for which the (*S*)-isomer was predicted to have a reversed orientation as compared to the binding modes of the (*S*)-stereoisomers of the other three compounds (see details in **Section 14** and Fig. S69, **Supplementary Information**).

We further conducted 500 ns molecular dynamics (MD) simulations for **5c**, **5d**, **5g**, and **5j** stereoisomers bound to VIM-2, respectively, to gain understanding of the dynamics of the enzyme-inhibitor complexes. As expected, the coordination of the phosphonate group and the Zn ions were robust over the entire simulation time and the fluctuation of ligand RMSD could be attributed to the movement of the aromatic moieties connected to

the amide bond (Supplementary Figs. S66 and 67). This observation suggests that those aromatic moieties lack strong interactions with the protein and are solvent exposed. The four ligands show conserved binding pattern to VIM-2.

Having validated the computational method against the X-ray data, we docked inhibitors **5a, b, e, f, h, k–m** to VIM-2 as well as **5a–m** to NDM-1 and GIM-1 (Supplementary Fig. S63). These computations reveal similar binding modes for all inhibitor—enzyme complexes. Thus, the coordination of the phosphonic acid moiety of the inhibitors to the Zn ions of the enzyme active site is conserved, and is the key interaction (Supplementary Figs. S63 and S68). The computationally predicted binding modes are in agreement with the chemical shift changes observed at NMR titrations of NDM-1. Hence, upon titration of compound **5f** to NDM-1, we observed significant chemical shift perturbations (chemical shift perturbations were considered significant above the population mean plus at least one standard deviation[56,57]) of His122, Asp124, Gly188, Thr190, Ser191 (His118, Asp120, Gly195, Thr197, Ser198, BBL)[34], which are all residues close to the Zn coordination sites of the enzyme. Furthermore, in agreement with the structures predicted by computational docking, amino acids Asp66, Met67 of loop 3, Val73, Ser75, Asn76 of β-sheet 3, Thr91, Trp93 of loop 5, and Lys211 of loop 10 (Asp60, Met61, Val67, Ser69, Asn70, Thr85, Trp87, and Lys224, BBL)[34] showed significant chemical shift perturbation, caused by their direct involvement into protein binding or due to binding induced conformational changes. Amino acids Asp66, Trp93, Asp124, Ser191, and Lys211 (Asp60, Trp87, Asp120, Ser198, and Lys224, BBL)[34] of NDM-1 have previously been reported to be important for inhibitor binding[22,26,28,58,59]. The docked poses for all (*R*)-isomers are highly similar, and they are similar to the binding modes of these compounds to VIM-2. The computationally docked binding poses of the (*S*)-isomers of these inhibitors are also similar, and rotated with ~180° as compared to the (*R*)-isomers, which is in agreement with that observed for VIM-2 binding (Fig. 5 and Supplementary Fig. S63).

The binding mode of **5a–m** to GIM-1 is predicted to be similar to the experimentally confirmed binding modes of the same inhibitors to VIM-2 and NDM-1. Some of the amino acid residues responsible for Zn coordination (His118, His196, Asp120, His263, BBL)[34] and those around the hydrophobic loop 3, and of flexible loops 5 and −10 are predicted to be involved in inhibitor binding. For the (*R*)-isomers, the docking predicted similar binding poses, except for (*R*)-**5k** and (*R*)-**5m** (Supplementary Fig. S64). The poses for (*R*)-**5k** and (*R*)-**5m** are comparable, with the benzyl/benzothiophene moiety being oriented towards Trp228 (BBL numbering)[34] (Supplementary Fig. S65) giving rise to a possible weak π–π stacking interaction, while in all the other cases, the benzyl/benzothiophene moiety of the (*R*)-isomers are oriented towards His118 (BBL numbering)[34]. The binding pose for the phosphonic acid functionality to Zn is conserved, independent of the stereochemistry and the variable moiety of the inhibitor, and of the enzyme (NDM-1, VIM-2, GIM-1) (Fig. 5). Similar to that observed for VIM-2 and NDM-1 binding, the (*S*)-isomers bind in a binding mode that is rotated by ~180° as compared to the binding mode of the (*R*)-isomers.

Whereas the binding affinities of the inseparable enantiomers of **5a–m** cannot be measured independently, the observation that VIM-2 complexes of both enantiomers of **5c-d** were obtained when the protein was crystallized from a 1:1 mixture of the enantiomers suggests comparable affinity. This observation is corroborated by the computational docking which suggest binding of both enantiomers of **5a–m** to the VIM-2, NDM-1, and GIM-1 binding sites with a ~180° rotated orientation yet with comparable binding scores, and comparable number of weak interactions to the protein binding site. The enantiomers may thus have comparable affinity and inhibitory affinity.

**Stereodynamic inhibitors are predicted to counteract resistance upon point mutations**

The fact that the interconverting stereoisomers of **5a–m** bind metallo-β-lactamases in different binding poses may counteract drug resistance

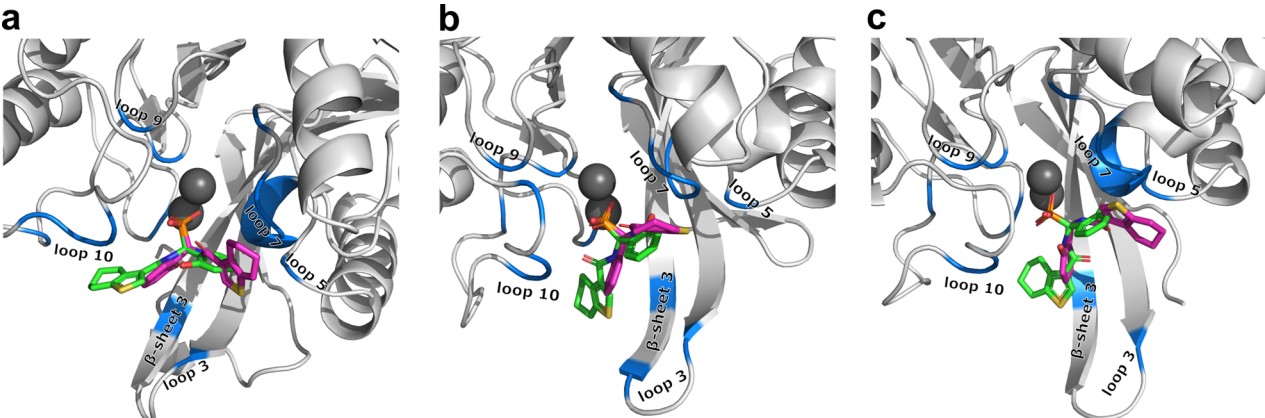

**Fig. 5 | The binding modes of the (*S*)- and (*R*)-isomers of the stereodynamic phosphonic acid-type inhibitors to the metallo-β-lactamases VIM-2, NDM-1 and GIM-1.** The computationally docked structures of the most potent broad spectrum inhibitors (*R*)-**5f** (magenta) and (*S*)-**5f** (green) bound to (**a**) VIM-2 (IC$_{50}$ = 48 μM) (**b**) NDM-1 (IC$_{50}$ = 30 μM) (**c**) GIM-1 (IC$_{50}$ = 52 μM). Residues oriented towards the bound ligand are highlighted with blue. The docking results show the conserved binding pattern in the same binding pocket for all the three enzymes. The orientation of the key functional groups is conserved, whereas the orientation of the ben-zothiophene moiety shows flexibility.

induced by single-point mutations. In order to prove this, we conducted computational saturation mutagenesis screening on the amino acids located in the binding pocket of VIM-2, NDM-1 and GIM-1 (Supplementary Tables S32–35) followed by molecular docking of both stereoisomers of inhibitors **5c** and **5d** to each mutant. The inhibitor binding affinities of 380 VIM-2 mutants were estimated using molecular docking, of which 37 reduced the binding affinity of the (*R*)-**5c** enantiomer, 105 of the (*S*)-**5c** enantiomer, and only 21 combinations affected the binding of both enantiomers (Fig. 6a). Hence, 94.5% of the mutations do not significantly weaken the binding of **5c**. For **5d**, 26 mutation combinations decreased the binding affinity of the (*R*)-enantiomer, 232 the binding of the (*S*)-enantiomer, and 19 combinations weakened the binding of both (Fig. 6b). Thus, 95.0% of the mutations do not weaken the binding of **5d**. Computational saturation mutagenesis followed by molecular docking experiments for the inhibitors **5c**, **5d**, **5g**, and **5j** to NDM-1 and GIM-1, and for **5g** and **5j** to VIM-2 (Supplementary Figs. S70–72) lead to the same conclusion as outlined for **5c** and **5d** when binding to VIM-2 mutants above. This observation suggests that the stereodynamics of the studied inhibitors decrease the risk for drug resistance development. Thus, specific single-point mutations of the target enzyme may weaken the binding of one of the enantiomers without affecting that of the other stereoisomer. The two stereoisomers rapidly interconvert in solution, and accordingly the active stereoisomer will be selected and enriched upon binding to the mutated enzyme. Hence, stereodynamics provide a greater adaptability for mutations for the inhibitors and thereby counter drug resistance induced by single-point mutations. It should, however, be mentioned that the greater adaptability of stereodynamic inhibitors might increase the risk of their binding to human metallo-enzymes and consequent unwished side effects. A higher risk of adverse off-target interactions is expected for compounds that sequester the zinc ion(s) of the metallo-β-lactamase binding site, such as EDTA and further chelators, whereas inhibitors specifically designed to bind zinc without chelation, such as taniborbactam, generally avoid human metalloenzymes.

## Conclusion

Developing novel strategies for metallo-β-lactamase inhibition is a crucial step in overcoming the challenges posed by antibiotic resistance. In this study, we introduce an approach utilising stereodynamic phosphonic acids for the inhibition of metallo-β-lactamases. Both interconverting enantiomers of this compound class bind the target bacterial enzyme, which is a rare feature in drug discovery. Our X-ray and computational investigations reveal that the two stereoisomers bind with different binding modes to the enzyme active site. Due to their unique adaptability, they are active against various metallo-β-lactamases including the clinically highly relevant NDM-

1 and VIM-2, and against GIM-1 that is the structurally most different member of the B1 metallo-β-lactamase enzyme family. Their structural adaptability is expected to counteract resistance development mediated by single point mutations as their binding does not depend on high affinity interaction to any specific amino acid. We demonstrate that this inhibitor class is easy to synthesize, soluble in water, membrane permeable, and nontoxic. They not just inhibit a series of metallo-β-lactamases in vitro, but also show activity on living bacteria.

The phosphonic acid moiety is shown to strongly promote binding to the Zn of the active site of metallo-β-lactamases and its introduction into small molecule inhibitors appears benefitial as compared to α-aminophosphonates and phosphonamidates that have previously been explored as transition state mimics.

Our results open the door for the exploration of the potential of stereodynamic inhibitors in other contexts, with the largest advantages being expected in the treatment of viral, fungal and bacterial diseases and of cancer, where resistance development due to single point mutations is a major challenge. Due to their geometric adaptability, fluxionally chiral compounds may, however, also find application in further fields of drug discovery and development, where structural adjustability may play a beneficial role.

## Methods
### General methods

Reagents, reactants, solvents and drying agents for synthesis, work-up and purification were purchased from commercial vendors (Sigma-Aldrich, VWR, Merck, Fluorochem, Thermo Fisher Scientific, Solveco), and if not stated otherwise they were used without further purification. Reactions and long term stability of the final compounds in buffer were monitored using LC-MS (Agilent 1260 Infinity II) equipped with an ESI-Q detector (Infinitylab LC-MSD) using Agilent InfinityLab Poroshell 120 C18 columns (2.7 μm, ϕ 2.1 mm L 50 mm or 2.7 μm, ϕ 4.6 mm L 50 mm). The intermediates were purified using VWR silica gel 60 (40–63 μm) with a Biotage Isolera One flash column chromatography instrument. The final products were purified with preparative RP-HPLC (VWR LaPrep P110) with single wavelength detection (254 nm), using a Kinetex C8 column (5 μm, 100 Å, ϕ 21.2 mm L 250 mm) and gradients of CH$_3$CN/H$_2$O + 0.1% formic acid as the mobile phase at a 10 mL/min flow rate. Analytical HPLC data were acquired on a VWR LaChrome ELITE system using an ACE3 C8 (3 μm, 300 Å, ϕ 3.0 mm L 100 mm) or a Kinetex C18 columns (5 μm, 100 Å, ϕ 3.0 mm L 150 mm). NMR spectra were recorded on a Varian Agilent MR400-DD2 400 MHz spectrometer equipped with a OneNMR probe, Bruker Avance Neo 500 MHz spectrometer equipped with a cryogenic TXO

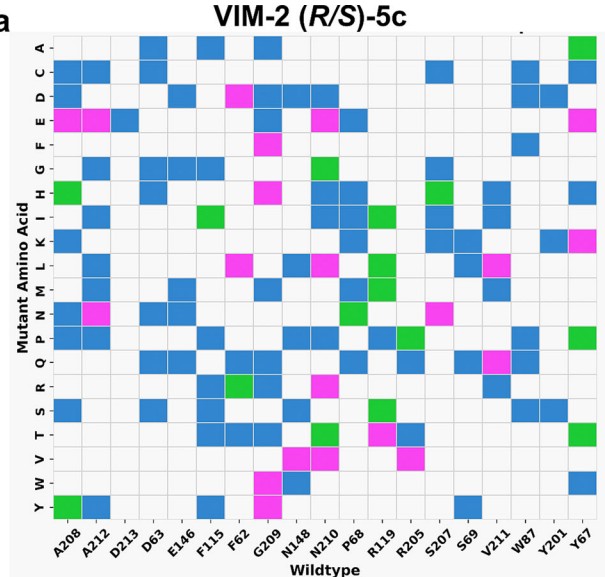

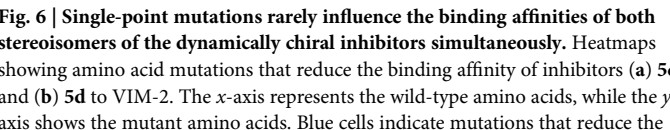

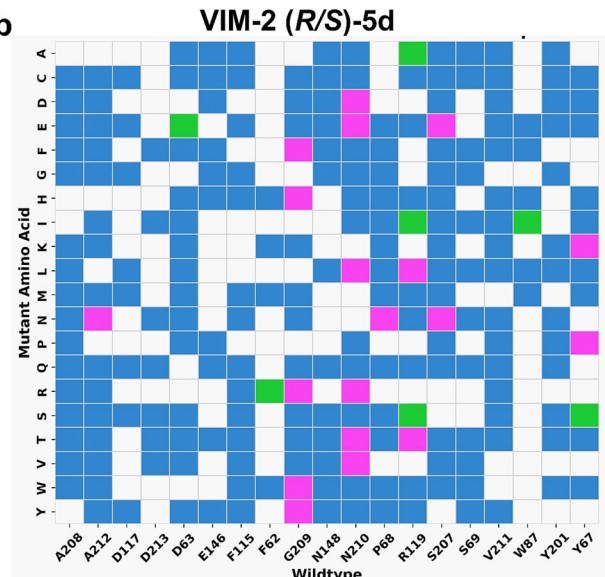

**Fig. 6 | Single-point mutations rarely influence the binding affinities of both stereoisomers of the dynamically chiral inhibitors simultaneously.** Heatmaps showing amino acid mutations that reduce the binding affinity of inhibitors (**a**) **5c** and (**b**) **5d** to VIM-2. The *x*-axis represents the wild-type amino acids, while the *y*-axis shows the mutant amino acids. Blue cells indicate mutations that reduce the binding of the (*S*)-enantiomer, green cells represent those that weaken the binding of the (*R*)-enantiomer, and pink-highlighted cells denote mutations that reduce the binding of both enantiomers. Since H240, C198, D118, H179, H116, and H114 of VIM-2 are key amino acids for Zn(II) ion binding, mutations in these amino acids were not considered.

probe, or a Bruker Avance Neo 600 MHz spectrometer equipped with a TCI cryogenic probe. The chemical shifts are reported using the residual solvent signal as an indirect reference to TMS.

The synthetic route towards compounds **5a–m** is summarized on Fig. 1b, and the details are given below. The synthesis towards the final compounds starts with the Kabachnik-Fields reaction, which creates the crucial phosphonate motif together with an amino group in one-step, where the latter is needed to synthetize a variety of amides. For the Kabachnik-Fields reaction of 2-methoxybenzaldehyde, diethyl phosphite as the phosphorous-containing reagent and hafnium-chloride as the catalyst in absolute ethanol at 60 °C were found to give the desired product in appropriate yield (73%) in 24 h. The benzyl deprotection of the secondary amine was conducted via a hydrogenation reaction using $H_2$ under atmospheric pressure and $Pd(OH)_2/C$ (Pearlman's catalyst) in ethanol in 19 h giving the common intermediate primary amine for the divergent synthetic strategy of the final amides. The amide coupling reactions were carried out either utilizing HATU or COMU as coupling agents with very similar success to activate the 13 carboxylic acids. DIPEA was used as a base and ethyl acetate as a solvent. The amides were synthetized in 24 h at room temperature giving moderate to good yields (61–95%) depending on the carboxylic acid. The last step was the ethyl ester hydrolysis parallel with the methyl ether bond cleavage with the use of boron tribromide in toluene within 7 h. The final products were obtained after preparative HPLC purification to >95% purity (determined by HPLC analysis, see **Section S3, Supplementary Information**) in yields of 46–86%. The final compounds showed water solubility and long-term stability (see **Section 4, Supplementary Information**). 2-Methoxybenzaldehyde can be prepared by O-methylation from 2-hydroxybenzaldehyde with MeI in the presence of NaH in a mixture of DMF and THF in good yield (94%) within 14 h (**Section 2, Supplementary Information**).

Detailed synthetic procedures for each reaction step are given in **Section 2, Supplementary Information**.

### Enzymatic metallo-β-lactamase assays
Half-maximal inhibitory concentration ($IC_{50}$) towards VIM-2, NDM-1 and GIM-1 were determined by testing twelve concentrations of inhibitor

compounds ranging between 0 and 800 µM. A few inhibitors were tested up to 1600 µM to ensure nearly complete inhibition. Solutions of 100 µL consisting of buffer, purified enzyme (1 nM for GIM-1 and VIM-2, 5 nM for NDM-1), inhibitor and reporter substrate nitrocefin (15 µM for VIM-2, 2 µM for GIM-1) or imipenem (30 µM for NDM-1) were incubated in triplicates on a 96 well plate for 5 min at 25 °C. Buffer solution consisted of 50 mM HEPES pH 7.2, 50 µM $ZnSO_4$ and 0.4 mg/ml bovine serum albumine. Inhibitors were dissolved in DMSO but diluted to assure <2% DMSO in measured assay. Absorbance was measured at 482 nm (nitrocefin) and 300 nm (imipenem) on a SpectraMax M4 spectrophotometer. The initial velocity at each inhibitor concentration was analyzed in GraphPad Prism v9.3.1 and fitted to dose-response curves by non-linear regression (log [inhibitor] vs response) used to calculate $IC_{50}$ (Supplementary Tables S2–5).

### Enzyme kinetic studies
Enzyme steady state kinetic studies were performed to determine the mechanism of inhibition using inhibitor **5g** and VIM-2 (**Section 7, Supplementary Information**). $K_m$', $K_i$, and $V_{max}$ values were determined (Table S6) by testing eight different nitrocefin concentrations (0–75 µM) both in the absence of inhibitor and in the presence of two inhibitor concentrations (36 µM and 72 µM). Assays were conducted in triplicates on a 96 well plate. Absorbance was measured at 482 nm using a SpectraMax 190 plate reader. Solutions of 300 µL consisting of buffer, purified enzyme (0.3 nM VIM-2), inhibitor **5g** and reporter substrate nitrocefin with final 2.5% DMSO were incubated in triplicates on a 96 well plate for 5 min at 30 °C. Buffer solution consisted of 50 mM HEPES pH 7.2, 50 µM $ZnSO_4$ and 0.4 mg/ml bovine serum albumin.

### Assessment of cytotoxicity using MTT assay
Cytotoxic activity of the compounds was measured through the MTT colorimetric assay as previously described[60] using the Colorimetric (MTT) Kit for Cell Survival and Proliferation (Merck/Millipore; Darmstadt, Germany). HepG2 cells (human hepatocellular carcinoma cell line) were kept in penicillin/streptomycin-free medium overnight and then they were treated with the compounds for 24 h. Subsequently, the MTT reagent was added to the cells and after 3 h of incubation, MTT-derived formazan crystals were

solubilized in acidified isopropanol (0.1 mL isopropanol with 0.04 N HCl). Absorbance was measured with multiscan spectrophotometer (Thermo Fisher Scientific) at 530 nm test and 630 nm reference wavelengths. Cell viability was expressed as the percentage of viable, untreated cells. $CC_{50}$ values were determined with GraphPad Prism software (v9.3.1) and adjusted to dose-response curves by non-linear regression (Supplementary Table S7).

## Cloning

The full-length $bla_{VIM-2}$, $bla_{NDM-1}$, and $bla_{GIM-1}$ genes, respectively, were subcloned between the NdeI and XhoI restriction sites of a modified pET28a vector, of which the T7 promoter was replaced with a tac promoter, resulting in plasmids pY326, pY296, and pY340.

## Membrane permeability

Preparation of OMVs. OMVs carrying active beta-lactamases VIM-2, NDM-1, or GIM-1 inside their lumen were prepared as described[46]. Briefly, *E. coli* strain BL21(DE3)ΔompA[61] carrying plasmid pY326, pY296, or pY340 were grown at 37 °C in LB media supplemented with 10 µM $ZnCl_2$ and 50 µg/mL Kanamycin. When cultures reached an optical density (600 nm) of 0.4, protein expression was induced by addition of 20 µM IPTG (cells carrying pY326) or 10 µM IPTG (cells carrying pY296 or pY340) and growth was maintained for 2 h. Cells were removed by centrifugation at $10,000 \times g$ for 10 min and cleared supernatants were filtered using 0.45 µm bottle top filters. OMVs were collected by centrifugation at $38,400 \times g$ for 2 h, washed in buffer (Dulbecco's PBS, pH 7.4, 8.1 mM $Na_2HPO_4$, 1.5 mM $KH_2PO_4$, 136.9 mM NaCl, 2.7 mM KCl, supplemented with 0.9 mM $CaCl_2$, 0.5 mM $MgCl_2$, and 10 µM $ZnSO_4$), filtered using 0.45 µm syringe filters, recollected by centrifugation at $74,500 \times g$ for 45 min and resuspended in buffer. The total protein concentration was determined spectroscopically using spectra corrected for the light scattering contribution of the OMVs and assuming 1 Abs = 1 mg/ml as described[46].

Determination of in situ beta-lactamase activity in OMVs. The activity of beta-lactamases encapsulated in OMVs in presence of the compounds, L-Captopril (Sigma Aldrich), or buffer was determined by monitoring the hydrolysis of nitrocefin in clear flat bottom half area 96 well microplates (Corning 3696) by measuring the increase in absorbance at 500 nm on a Fluostar Omega plate reader (BMG Lab Tech) for 10 minutes. The concentration of the OMV suspensions was adjusted to 12.3 µg/mL for VIM-2 containing OMVs, 50.4 µg/mL for NDM-1 containing OMVs, and 48.9 µg/mL for GIM-1 containing OMVs with buffer to match the hydrolysis rate of the different suspensions. 50 µL of OMV suspension were incubated with 1 µL of compound (10 mM in $H_2O$) for 5 min prior to the addition of 50 µL nitrocefin solution (0.4 mM (40 µg/mL) in buffer). For each compound, the relative activity was determined based on the hydrolysis rate of the first 10 min of the reaction as $(\partial Abs/\partial t)_{compound}/(\partial Abs/\partial t)_{buffer}$. For each condition, four independent replicates were measured.

## Metallo-β-lactamase inhibitory activity on clinical isolates

Antibiotic susceptibility testing. Susceptibility was tested by agar disc diffusion, employing a EUCAST reference protocol[62]. Antibiotics containing discs (ampicillin-sulbactam [10 µg/10 µg], piperacillin-tazobactam [30 µg/6 µg], ceftazidim [10 µg] and meropenem [10 µg]) were obtained from Becton-Dickinson (BD, Heidelberg, Germany). The discs were supplemented with inhibitors (10 µg each) and the diameters of the inhibition zones were measured following overnight incubation. Wild-type *E. coli* TOP10 and unsupplemented discs were used as controls. *E. coli* TOP10 is a common *E. coli* lab strain allowing for high transformation efficiency (genotype F-mcrA D (mrr-hsdRMS-mcrBC) F80lacZDM15; DlacX74-recA1 deoR araD139 D(ara-leu)7697 galU galK rpsL(StrR) endA1 nupG). Figure 2d shows data points corresponding to the mean from two biological replicates, where each biological replicate represents the mean of two technical replicates. For wild type and VIM expressing *E. coli* with inhibitor, three biological replicates were acquired.

## Cloning of $bla_{VIM-2}$

For the expression of $bla_{VIM-2}$ in *E. coli* TOP10, the VIM-2 coding sequence was amplified from *P. aeruginosa* PA-A chromosomal DNA (PMID: 37379881) using primers VIM_for (5′- GTT GTG CCA AGT GCG CG -3′) and VIM_rev (5′- GCA GTA CCA CCC GAC AAT CTG -3′), encompassing the coding region and the expected native promoter. Amplicons were cloned into pCR4 (Invitrogene) using the TOPO TA cloning kit (Invitrogene) according to the manufacturer's protocol. *E. coli* transformants were selected on LB agar containing ampicillin (50 mg/ml). The correctness of the insert was verified by sequencing. The plasmid is designated pCR4-VIM-2. *E. coli* × pCR4-VIM-2 showed resistance to ampicillin-sulbactam (BD), piperacillin-tazobactam (BD) and reduced susceptibility to ceftazidim (BD). Susceptibility to meropenem (BD) remained unchanged compared to the wild type.

## $^{1}$H, $^{15}$N-HSQC titration experiments

The $^{1}$H, $^{15}$N-HSQC spectra for the titration experiments were recorded on a Bruker Avance Neo 600 MHz spectrometer equipped with a 5 mm TCI cryogenic probe at 25 °C. The U-[$^{15}$N]-labeled VIM-2 was prepared in a buffer of 20 mM $KPO_4$, 0.1 mM $ZnCl_2$, at pH 7.0, containing 85% $H_2O$/15% $D_2O$ at a concentration of 0.25 mM. The U-[$^{15}$N]-labeled NDM-1 was used in a buffer of 20 mM HEPES, 0.1 mM $ZnCl_2$, at pH 7.0, containing 85% $H_2O$/15% $D_2O$ at a concentration of 0.25 mM. The ligands **5d**, **5g**, **5i**, **5j** (for VIM-2 titrations) and **5f** (for NDM-1 titration) were dissolved in the same buffer as the corresponding enzymes. Spectra were recorded using solvent suppression, 1024 points in f1, 16 transients, 512 increments in f2 and a relaxation delay of 0.2 s. The titrations were carried out with 0, 0.25, 0.50, 1.00, 1.50, 2.25, 3.00, 4.00, 5.00, 7.00, 10.0 equivalents of the inhibitors.

## VIM-2 inhibitor co-crystallization

Crystallization conditions were adapted from Christopeit et al.[63]. VIM-2 crystallization was performed in MRC maxi 48 well sitting-drop crystallization plates (Swissci). For native VIM-2 crystallization, 1.5 µl protein solution was mixed (10.3 mg/ml in 50 mM Tris/HCl, pH 7.2, 100 µM $ZnCl_2$) with 1.5 µl precipitant solution (25% (w/v) polyethylene glycol (PEG) 3350, 0.2 M magnesium formate). The crystallization cocktail was placed over 200 µl of the precipitant solution in the reservoir. Crystals (Supplementary Fig. S50) grew within 3 days and showed good diffraction properties. To support the nucleation in co-crystallization approaches, a native VIM-2 crystal was crushed, and a seed stock was prepared by vortexing the crushed crystal with 50 µl reservoir solution and 3 glass beads in a reaction tube for 6 × 30 s. For co-crystallization approaches, each inhibitor (compound **5d**, **5c**, **5j**, and **5g**) was dissolved in the crystallization solution (25% (w/v) polyethylene glycol (PEG) 3350, 0.2 M magnesium formate) at 5 mM, 10 mM, and 20 mM concentrations, respectively. Next, 1.5 µl crystallization solution was mixed with 1.5 µl purified VIM-2 protein solution (10.3 mg/ml in 50 mM Tris/HCl, pH 7.2, 100 µM $ZnCl_2$) and VIM-2 crystal seeds were introduced via streak seeding. Crystals were left to grow and were fished after 14 days, soaked in cryo protectant solution (20 mM inhibitor, 25% (w/v) PEG3350, 15% (v/v) ethylene glycol, 0.2 mM $MgCl_2$, 50 mM HEPES pH 7.2) and flash frozen in liquid nitrogen.

## Reporting summary

Further information on research design is available in the Nature Portfolio Reporting Summary linked to this article.

## Data availability

The datasets generated during and/or analysed during the current study (FIDs, crystallographic CIF files, computed structures of the enzyme-inhibitor complexes (pdb)) are available in the Zenodo repository, DOI:10.5281/zenodo.12571911. The atomic coordinates and experimental data for the X-ray structures of the co-crystal of VIM-2 with **5c** (9F0P), with **5d** (9F0Q), with **5g** (9F0R) and with **5j** (9F0S) have been deposited to the Protein Data Bank (www.pdb.org). Synthetic procedures, NMR spectra and analytical data for the synthesized compounds,

details of the X-ray crystallographic analyses and the cytotoxicity, enzyme inhibition, membrane permeability and bacterial assays are given as **Supplementary Information**.

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

## Acknowledgements

The Lund Protein Production Platform (LP3) at Lund University, managed by Wolfgang Knecht, is acknowledged for NDM-1, GIM-1 and VIM-2 protein expression. This project made use of the NMR Uppsala infrastructure, which is funded by the Department of Chemistry—BMC and the Disciplinary Domain of Medicine and Pharmacy, Uppsala University. Computations were performed on resources provided by Swedish National Infrastructure for Computing (SNIC) through National Supercomputer Center (NSC) under Project NAISS 2023/5-392 and 2024/5-583. The synchrotron data were collected at beamline P14 operated by EMBL Hamburg at the PETRA III storage ring (DESY, Hamburg, Germany). We would like to thank Selina Storm for the assistance in using the beamline, Ruisheng Xiong for assistance with HRMS analyses, and Hanna Andersson for helpful discussions in the initial phase of this project. We acknowledge the Swedish Research Council (2013-8804, 2024-05496), NSFC-STINT (82211530060), and the Uppsala Antibiotic Center for financial support. J.T. was supported by the Swedish Society for Medical Research (PD20-0191) and the Åke Wiberg Foundation (M22-0199). E.C.S gratefully acknowledges support by the Max Planck Society, the DFG via grant No. 458246365, and by the Federal Ministry of Education and Research, Germany, under grant number 01KI2114. Project no. TKP2021-EGA-24 was implemented with the support provided by the Ministry of Innovation and Technology of Hungary from the National Research, Development and Innovation Fund, and financed under the TKP2021-EGA funding scheme.

## Author contributions

K.V.G. synthesized, purified and analyzed the inhibitors, and did the NMR binding studies, the data analysis and prepared figures. F.D. contributed to the development of synthetic routes. A.A.R. and K.V.G. expressed and purified the $^{15}$N-labelled enzymes. D.S. and H.K.S. performed the IC$_{50}$ enzymatic assays. K.V.G. and M.W. carried out the Lineweaver-Burk analysis. V.T. and M.Cs. measured the cytotoxicity. L.Z., Z.X., and W.Z. performed the molecular docking and dynamics simulations. J.L. performed the computational studies on the possible resistance development. J.T. performed the membrane permeability studies. K.B. and A.P., expressed, purified and co-crystallized VIM-2 with the inhibitors and collected the data and prepared the figures. K.B., R.B., and E.C.S solved the structures and analyzed the data. P.H. and H.R. performed enzyme inhibitory activity measurements on living bacteria. M.E. conceived and supervised the project. All authors contributed to the writing.

## Funding

## Competing interests

The authors declare no competing interests.
