## [Transparent Peer Review file · Communications Chemistry]

Dynamically chiral phosphonic acid-type metallo- β -lactamase inhibitors

Corresponding Author: Professor Mate Erdelyi

Version 0:

Reviewer comments:

Reviewer #1

(Remarks to the Author)

The manuscript by Virág Gulyás reports on the design, synthesis and characterization of 13 original amino-phosphonate derivatives, showing inhibitory activity on metallo-beta-lactamases, including the epidemiologically relevant NDM-1 and VIM-2 enzymes.

The conceptual approach, i.e. relying on dynamically chiral compounds, appears to be brilliant and its application to metallo-beta-lactamase inhibition is original. The authors provide convincing data (even somewhat too redundant) regarding the binding of the two stereoisomers in different metallo-beta-lactamases. However, the manuscript does not provide a convincing demonstration of the actual advantage of such stereodynamic compounds to address resistance (this part of the manuscript being extremely speculative and not supported by any experimental data, see below – major comments). Furthermore, some key results are not adequately presented, limiting the independent assessment of the actual inhibitory potency and synergistic activity of the tested compounds (see major comments below), further complicated by potential inaccuracies in the Methods section.

Overall, the manuscript discloses promising results and very interesting and comprehensive structural data, but the part dealing with the biological characterization of the inhibitors should be significantly improved to be acceptable for publication in Communications Chemistry.

Major comments

1 – Enzymes assays and inhibitory activity.

Figure 2a. Reporting IC₅₀ values does not allow to really assess the actual inhibitory activity of the compounds, as it depends on the experimental conditions (reporter substrate and K_m value). The authors should provide proper dissociation constants, and ideally confirm the (likely) competitive mechanism of inhibition. Such K_d values are easily computed from available data (refitting data using $v_0/v_i = f([I])$ plots, or applying the Cheng-Prusoff equation once competitive inhibition is confirmed). Using the latter method and the information provided in the Methods section, compound 5g would show an approximate K_i value of 6 μ M, which is, to be honest, rather high. It would be useful to compare and discuss the potency of these compounds with other structurally related inhibitors, including from the same group (see e.g. references 22, 25 and 27). The K_i values of a commercially available reference compound (e.g. captopril, taniborbactam, etc.) could be included here.

Lines 345-355. The Methods section apparently contains some mistakes that further complicates the assessment of the results described in the manuscript:

(a) enzymes resuspended in DMSO???

(b) enzyme assay with 2 μ M nitrocefin with GIM-1 (in their setup the maximum variation of absorbance upon complete substrate hydrolysis would be extremely low)

(c) some enzymes are rather sensitive to the presence of low concentrations of DMSO, did the authors check the absence of inhibition with 2% DMSO in the reaction mixture?

(d) imipenem hydrolysis was followed at 380 nm, which is an unusual wavelength for this substrate (is imipenem even absorbing at this wavelength?), what was the variation of extinction coefficient upon substrate hydrolysis at this wavelength?

Lines 145-155. This sentence (lines 149-150) seems awkward and raise the question about the actual selectivity of the inhibitors, which is not sufficiently discussed in the manuscript. The lack of cytotoxicity on specific cell lines does not necessarily correlate with other off target activities on human enzymes, for example the angiotensin-converting enzyme 1.

Testing the inhibitory activity of the compounds on that enzyme or another human metallo-hydrolase would contribute to increase the significance of the article.

2 – Microbiological assays

Figure 2c. The authors should report the actual diameters of the growth inhibition zones measured with and without the inhibitor. Indeed, mentioning a percentage of variation of this value does not allow to assess the initial level of resistance of the tested strain, nor the actual synergistic activity of the inhibitor.

Lines 397-399. The quantity of compound added to the discs should be mentioned here.

Lines 400-408. It is unclear why the authors cloned a beta-lactamase gene in a beta-lactamase encoding plasmid, considering the variety of commercially available plasmids with other resistance markers. Indeed, the authors have to use sulbactam or tazobactam containing discs to assess the contribution of VIM-2 production on the susceptibility phenotype of the strain. Please comment.

Lines 168-177. The title of this section is a strong assertion based on the available data. Indeed, only one recombinant (not a clinical isolate) strain was tested and no detailed susceptibility data are reported (see above). These data could be encouraging but should be put into context. Line 173, "in vivo" is not appropriate here. The authors mention 10 to 60% increase of the growth inhibition zone diameter, but Figure 2c does not show anything above roughly 25%. Please clarify. It should also be interesting to discuss the correlation between inhibitory potency and synergistic activity, especially considering the availability of inhibition data on OMVs. Indeed, compound 5e shows the highest potentiation while being a rather poor inhibitor.

Lines 156-167. Again, the conclusion seems to be strong based on the provided data. When compared to captopril, the level of phosphonate-mediated MBL inhibition in OMVs appears much lower. It is difficult to understand whether this is due to poor penetration or lower inhibition potency overall, because the IC₅₀ (or better the K_d) of captopril is not provided. It would be interesting to compare the level of inhibition in OMVs with the level of inhibition expected in similar conditions but with the free enzyme (again easily computed if the K_d is known). This would give a really interesting insight about the capability of the compounds to cross the outer membrane.

3 – Discussion and implications

The structural data are very convincing but are typically obtained in the presence of rather high concentrations of inhibitors. The authors interestingly observe the two enantiomers bound in the enzyme active site, but do not really discuss whether these two species could have similar inhibitory potencies or not (see e.g. lines 289-292).

The approach is very interesting, but the authors could seem a bit overenthusiastic regarding the fact that the stereodynamic compounds would mitigate resistance, even with a single amino acid substitution. Indeed, in the complexes showing both enantiomers, the interactions appear to apparently involve similar amino acids (Tyr67, Trp87 seem important in both complexes). Please discuss. It would have been very interesting to measure the inhibitory activity of one compound with additional NDM- and VIM-type variants. Such data, although not equivalent to the characterization of laboratory-selected resistant mutants, could have been supportive of such statement.

Lines 228-231. Could the relevant dynamic adaptability of the compounds, presented as an advantage here, be a limitation for the selectivity of the compounds (especially regarding human metallo-enzymes)? Please discuss.

Minor comments

Lines 44-46. This sentence is unclear. What are the "limited therapeutic options" intended by the authors? Polymyxins?

Lines 50-51. Both xeruboractam and taniboractam are in clinical development (phase 1 and phase 3, respectively). Reference 11 should be added,

Lines 56-57. This is well true but not limited to taniboractam. Almost all antibiotics had resistance strains/mechanisms identified prior to their approval, starting with penicillin.

Lines 77-78. It would be useful to remind the reader what are the structural peculiarities of GIM-1 that makes it so different from other subclass B1 enzymes.

Line 171. Please specify the recipient strain (TOP10?).

Lines 183-184. Very nice and convincing data and interpretation. Kinetic studies (e.g. Dixon plot) to confirm the competitive nature of the inhibition mechanism with a single compound could be an asset here.

Lines 205. The numbering used in the paper is confusing. Please use the consensus numbering scheme established for class B beta-lactamases throughout or at least provide the equivalence.

Figure 5. As such, not really useful (too small, absence of labelling).

Line 364. EC50 or CC50 instead of IC50?

Line 390. Please provide the molar concentration of the substrate.

Reviewer #2

(Remarks to the Author)
Review attached.

Reviewer #3

(Remarks to the Author)

This is a useful contribution, providing further data on phosphonates, which are well known metallo-enzyme inhibitors. The manuscript lacks comparisons and discussions with previous literature (supporting figures of comparisons needed, and discussion in the 'discussion' section). Some important details of inhibition are missing (see below). There is emphasis in the discussion on the importance of the stereochemistry, which is interesting but it is unclear how important this actually is to inhibition as the compounds are not purified as stereochemically pure, so some of the conclusions appear to be speculative and not fully demonstrated.

The work by Gulyas et al. describes the inhibition of three B1 MBLs by phosphonate inhibitors. Phosphonates are well known inhibitors of metallo-enzymes and metallo-beta-lactamases and the work presented here shows a new series of 13 phosphonates as μM inhibitors of three B1 MBLs. The structural data on their interactions with one, VIM-2, is interesting as it shows that the inhibitors can bind in a dual conformation, dependent on the stereochemistry. These data are worthy of publication, although there are some issues that should be addressed:

1. Importantly, there are not enough comparisons to previous work in this field. In particular, there should be comparisons against the structures described by Chen's group (<https://pubmed.ncbi.nlm.nih.gov/31483651/>), and Dmitrienko's group (<https://pubmed.ncbi.nlm.nih.gov/29485857/>), both in the text and in supporting figures. How does binding compare, which stereoisomers do they most closely resemble, what inhibition do they show? Also how does binding compare to hydrolysed antibiotics (comparison figure against, for example, the recent VIM-2:biapenem work, PDB 6Y6J)?
2. Related to this, there should also be discussion and a comparison of the binding of these phosphonates with taniborbactam to VIM-2. This is important because the authors state that binding mimics the transition state of β -lactam hydrolysis, which is how boronates, like taniborbactam, bind. The relationship to the transition state is interesting but is not demonstrated directly. The structural reasoning with the TS should be explained further and justified.
3. Some of the discussion of the different modes of binding for the different stereoisomers is not fully justified. In particular, it is unknown if this is important for binding different MBLs (there is no data to show the inhibition of each stereoisomer, because they are purified as a mixture). Also, there is no evidence that these could "counteract resistance development", as only 3 enzymes are tested.
4. Do the compounds exhibit any time-dependent inhibition? i.e. what happens to the IC50s when the protein is incubated for longer? The authors also mention no zinc chelation is observed in the NMR (line 184), but it would be useful here to say over what timescale this is (minutes, hours?). It is possible that zinc chelation may be time-dependent (over hours and not minutes).
5. For the structures of VIM-2 with 5j and 5g, the crystal table needs to show stats for the resolution range that was refined against (i.e. high resolution of 1.4 and 1.36, respectively).
6. For the IC50s the authors should explain how the standard error of the mean was calculated (it suggests at least 3 independent IC50 values were calculated – not just technical replicates – which is not mentioned in the methods), and why this is presented rather than the standard error of the logIC50, which is more accurate for non-linear regression.
7. S144, S145, S146 are barely readable due to their size and low resolution, but are important for understanding the quality of the crystallographic data. These should be split up so that the images are viewable and at least the electron density can be seen properly (particularly at the resolution in the PDF).
8. Line 51 clarify that it is the boronate hydroxy group that binds.
9. Include the ligand RSCCs (from PDB validation reports) in the text (including of both binding conformations), or as a supporting table.

Reviewer #4

(Remarks to the Author)

Referee report for:

Dynamically chiral phosphonic acid-type metallo- β -lactamase inhibitors

In this manuscript, the authors present the synthesis and biological activity of chiral α -amino phosphonic acid as metallo-beta lactamase inhibitors. The binding mode of selected inhibitor-enzyme complexes was evaluated by NMR and X-ray crystallography and molecular modeling. The inhibitors were active in enzymatic and bacterial assays with the best inhibitors showing activity in the low micromolar range, with no detectable cytotoxicity.

The main claim of the work is that the α -amino phosphonic acids designed by the authors can stereochemically adapt via

racemisation to the binding pocket of different enzymes and as such provide potentially broader enzyme coverage. The concept of stereochemically adaptable inhibitors is intriguing and relevant for further inhibitor design and should be considered novel in the context of MBL inhibition and small-molecular inhibitors.

However, before publication, the authors should clarify the following:

- All the inhibitors are prepared as racemic mixtures containing both enantiomers of the inhibitors. Can the authors be sure that the α -amino phosphonic acids are racemized i.e. are "dynamically chiral" under assay conditions? I cannot find documentation for the dynamic behavior in the manuscript. For comparison, amino acids in the acid form are not particularly prone to racemization under mild conditions (Bada, J.L., Kinetics of racemization of amino acids as a function of pH, J. Am. Chem. Soc. 94(4):1371–1373, 1972; Bada, J.L. and Schroeder, R.A., Amino acid racemization reactions and their geochemical implications, Naturwissenschaften 62:71–79, 1975).

- Line 88/89: The authors write "Once bound to the enzyme's active site, the phosphonic acid remains strongly coordinated to the Zn ions.....» does this imply that the inhibition is irreversible? Have the authors tested this by e.g. jump-dilution experiments?

Minor comments:

- * The authors use the terms "dynamically chiral" "stereodynamic binding" and "stereodynamic inhibitor" in the manuscript. Even though, there is a reference (line 105) in the text, the manuscript would benefit from defining the concept briefly in the text.

- * Line 288: "novel and innovative" should be rewritten.

The experimental section is carefully prepared and appears as reproducible.

Sincerely, Prof. Annette Bayer, UiT

Version 1:

Reviewer comments:

Reviewer #1

(Remarks to the Author)

The authors provided detailed answers to the referees' comments, some convincing, some a bit less (with also a few conceptual mistakes, such as in page 4, vaborbactam is not an MBL inhibitor). Disappointingly, many new and interesting data were included in the SI rather than in the manuscript. More importantly, the authors provided evidence to support the competitive inhibition of at least one compound (Fig. S161). Please note that in Table S35, what is reported as the K_m value (which does NOT vary) is actually the apparent K_m value (K_m app or K_m'). I still believe that the way the authors present their AST data is not appropriate (mentioning that the purpose of such experiments is "not to quantify resistance", one might then ask what "antimicrobial susceptibility testing" means). These data (Fig. 2C) are presented in an unconventional and misleading way, as the analysis of the raw data does not allow to strongly assert that "the compounds inhibit VIM-2 in bacteria", considering that very limited, if any, AMP/SUL potentiation is observed (in sharp contrast with "We agree that relative increases in inhibition zone diameter do not allow conclusions to be drawn about the baseline level of resistance" as stated in the rebuttal). The absence of significant synergistic activity is obviously not a problem for publication (as it is often the case for exploratory research) but data should be analyzed and conclusions drawn with objectivity. Also, "bacterial growth inhibition is influenced by a number of further factors, such as additional bacterial resistance mechanisms" might well be true but using a laboratory strain producing the MBL, one might wonder what these additional mechanisms could be. The addition of the K_i values (Table S2) show that the inhibitors are indeed not very potent, this might also be the reason underlying the lack of potentiation on *E. coli* cells. Overall, the inhibitor design and overall approach remains very interesting.

Reviewer #2

(Remarks to the Author)

The authors have addressed the comments of the previous report.

Reviewer #3

(Remarks to the Author)

The authors have improved the paper in the revised version and addressed my comments. Overall, I think this is a useful contribution and is good work. I do not think that it has quite the transformative potential that is suggested, but it is useful in this field.

Reviewer #4

(Remarks to the Author)

I am happy with the response by the authors to the questions raised by my colleagues and myself and have no further comments to the manuscript.

I recommend publication as is.

We would like to take this opportunity to express our gratitude to the peer reviewers for their constructive and highly valuable insights. We hope that the revised manuscript is acceptable for publication in *Communications Chemistry*.

Response to the comments of Referee-1:

The manuscript by Virág Gulyás reports on the design, synthesis and characterization of 13 original amino-phosphonate derivatives, showing inhibitory activity on metallo-beta-lactamases, including the epidemiologically relevant NDM-1 and VIM-2 enzymes.

The conceptual approach, i.e. relying on dynamically chiral compounds, appears to be brilliant and its application to metallo-beta-lactamase inhibition is original. The authors provide convincing data (even somewhat too redundant) regarding the binding of the two stereoisomers in different metallo-beta-lactamases. However, the manuscript does not provide a convincing demonstration of the actual advantage of such stereodynamic compounds to address resistance (this part of the manuscript being extremely speculative and not supported by any experimental data, see below – major comments). Furthermore, some key results are not adequately presented, limiting the independent assessment of the actual inhibitory potency and synergistic activity of the tested compounds (see major comments below), further complicated by potential inaccuracies in the Methods section.

Overall, the manuscript discloses promising results and very interesting and comprehensive structural data, but the part dealing with the biological characterization of the inhibitors should be significantly improved to be acceptable for publication in *Communications Chemistry*.

Major comments:

#1. – Enzymes assays and inhibitory activity.

Figure 2a. Reporting IC₅₀ values does not allow to really assess the actual inhibitory activity of the compounds, as it depends on the experimental conditions (reporter substrate and K_m value). The authors should provide proper dissociation constants, and ideally confirm the (likely) competitive mechanism of inhibition. Such K_d values are easily computed from available data (refitting data using $v_0/v_i = f([I])$ plots, or applying the Cheng-Prusoff equation once competitive inhibition is confirmed). Using the latter method and the information provided in the Methods section, compound 5g would show an approximate K_i value of 6 μM, which is, to be honest, rather high. It would be useful to compare and discuss the potency of these compounds with other structurally related inhibitors, including from the same group (see e.g. references 22, 25 and 27). The K_i values of a commercially available reference compound (e.g. captopril, taniborbactam, etc.) could be included here.

Response: We thank the Reviewer for the helpful suggestion. We included a discussion of the potency of the presented compounds in comparison to structurally related ones, as

suggested, into the main text and a supplementary Table and Figure into the supplementary information:

*"These enzyme inhibitory activities indicate that incorporating a phosphonic acid moiety into small molecule metallo- β -lactamase inhibitors enhances their potency when compared to compounds that have no phosphonic acid group, or have a carboxylic acid group at a comparable position.^{23,25} As phosphonates show affinity to metal ions, earlier approaches focused on combining a phosphonate group with other metal binding groups, such as thiols^{27,42} and carboxylic acids^{23,25,43} (Table S34, Supplementary Information). In another approach, the phosphonic acid moiety was incorporated into β -lactam resembling structures, resulting in β -phospholactams⁴⁴ or their chemically more stable open analogue piperidinyl α -aminophosphonates.²² The phosphonic acid-type inhibitors **5a-m** coordinate to the zinc ions, analogous to the previously reported phosphonates.^{22,23} The structurally closest analogue piperidine-ring containing α -aminophosphonates showed IC_{50} 4.1 - 328 μ M and K_d values 0.4 - 15 mM against VIM-2, and IC_{50} 7.9 - 506 μ M and K_d of 0.5 - 3.1 mM against NDM-1.²² β -Phospholactam, the analogue that best resembles a β -lactam so far, showed 53% inhibition of NDM-1 at 100 μ M concentration. The limited biological data available for this compound may be the consequence of demanding and low yielding synthesis, limited stability and low aqueous solubility. When connected to heteroaryl moiety, phosphonates show K_i 0.3 - 30.3 μ M against VIM-2, and 31.4 - 741.3 μ M against NDM-1.²³ Heteroaryl 6-phosphonomethylpyridine-2-carboxylates encompassing several zinc-binding motifs showed similar IC_{50} values (0.464 - 1.90 μ M) against VIM-2, and improved inhibition of NDM-1 (IC_{50} 0.306 - 0.374 μ M).²⁵ The fully aliphatic N-(phosphonomethyl)-iminodiacetic acid, in which the phosphonic acid group is connected to two carboxylic acids through a nitrogen, was among the most potent compounds with IC_{50} 0.68 μ M against VIM-2 and 0.91 μ M against NDM-1. It showed synergistic effect when combined with meropenem, and a high affinity to zinc ions (K_d of 56 nM) due to zinc sequestration.²⁵ Furthermore, mercaptoethylphosphonates were presented as metallo- β -lactamase inhibitors with IC_{50} 0.38 - 133 μ M against VIM-2, 1.8 - 144 μ M against NDM-1, and 0.18 - >5000 μ M against GIM-1.²⁷ X-ray crystallography indicated that only one of the stereoisomers of chiral mercaptophosphonates bind VIM-2, by coordination of its sulfur to the zinc ion of the enzyme whereas its phosphonate group interacts with the sidechain of Asn233.²⁷ Overall, phosphonic acids and their close derivatives show high potential for use as metallo- β -lactamase inhibitors. Incorporation of a phosphonic acid moiety provides μ M potency against metallo- β -lactamases, which may be improved by further protein-binding moieties. Accordingly, phosphonates are a promising starting point for drug development."*

There to, we have added a figure showing the superimposition of the binding mode of the presented inhibitors with that of previously developed phosphonates (Figure S150).

The IC_{50} values for commercially available inhibitors have been added to the text ("Two inhibitor candidates are in clinical trials currently, xeruborbactam (IC_{50} 0.1 μ M for VIM-2; 4.3 μ M for NDM-1) and taniborbactam (IC_{50} 0.04 μ M for VIM-2; 0.1 μ M for NDM-1), which are

both bicyclic boronates.^{11,19-21} and further down "L-captopril (IC₅₀ 4.4 μM)⁴⁷". We could not find K_i values for these compounds, as IC₅₀ values are most commonly reported in the medicinal chemistry, not K_is.

Competitive inhibition was confirmed by additional Lineweaver-Burk analysis of the inhibition of VIM-2 by compound **5g**. This data has been added to the supplementary information, and the conclusion to the main text as follows: "A Lineweaver-Burk analysis⁴¹ confirmed competitive inhibition (**Figure S161, Supplementary Information**)."

Estimated inhibition constants have been added to the supplementary information (Table S2), as requested, and a sentence has been added to the main text as follows: "Estimated inhibition constant (K_i) for VIM-2, GIM-1 and NDM-1 are given in Table S2, Supplementary Information."

#2. Lines 345-355. The Methods section apparently contains some mistakes that further complicates the assessment of the results described in the manuscript:

(a) enzymes resuspended in DMSO???

(b) enzyme assay with 2 μM nitrocefin with GIM-1 (in their setup the maximum variation of absorbance upon complete substrate hydrolysis would be extremely low)

(c) some enzymes are rather sensitive to the presence of low concentrations of DMSO, did the authors check the absence of inhibition with 2% DMSO in the reaction mixture?

(d) imipenem hydrolysis was followed at 380 nm, which is an unusual wavelength for this substrate (is imipenem even absorbing at this wavelength?), what was the variation of extinction coefficient upon substrate hydrolysis at this wavelength?

Response:

Lines 345-355a+d: The Method section has been adjusted to accurately reflect the experiments. Inhibitors were dissolved in DMSO and diluted to <2%, not the enzymes. Imipenem was tested at a wavelength of 300 nm, not 380 nm as previously stated.

Lines 345-355b: The amount of substrate used for each enzyme was determined based on K_m values. Since the substrate in the inhibitor assay should be below K_m, we were limited to test at this range. The absorbance variation at this substrate concentration is low and not ideal, but we still managed to produce reliable values.

Lines 345-355c: Concentrations at <2% DMSO are generally considered unlikely to affect activity in any significant way, unless the enzymes are specifically sensitive to organic solvents. The use of 2-3% in metallo-beta-lactamase studies is common and was validated by Skagseth *S. J. Med. Chem.* 2017, 135, 159-173 that DMSO does not inhibit metallo-beta-lactamases at this concentration. We followed this procedure in our previous publications as well, for instance in *RSC. Med. Chem.* 2023, 14, 2277, *ACS Omega* 2022, 7, 4550.

#3. Lines 145-155. This sentence (lines 149-150) seems awkward and raise the question about the actual selectivity of the inhibitors, which is not sufficiently discussed in the manuscript. The lack of cytotoxicity on specific cell lines does not necessarily correlate with other off target activities on human enzymes, for example the angiotensin-converting enzyme 1. Testing the inhibitory activity of the compounds on that enzyme or another human metallo-hydrolase would contribute to increase the significance of the article.

Response: We have rephrased the misleading sentence to "*No obvious correlation between structure of the amide attached moiety and the enzyme inhibitory activity was observed*"

We thank the Referee for the suggestion of studying the inhibition of human metallo-enzymes. We have not had access to assays of ACE-1 and other human metallo-enzymes so far. Upon this helpful comment, we will attempt to get such data for our upcoming papers, even if this is unrealistic to achieve for the current compounds.

When searching the literature, we noted that no data for off target activities on human enzymes is available for most of the published inhibitors, for instance for taniborbactam and vaborbactam. These also bind the zinc ions of metallo-beta-lactamases and currently are in clinical development (taniborbactam finished Phase 3). We noted that a higher risk of adverse off-target interactions was reported for compounds that sequester the zinc ions of the binding site (EDTA and other broad-spectrum chelators), whereas inhibitors specifically designed to bind zinc without chelation, such as taniborbactam, generally avoid human metalloenzymes. This certainly does not confirm that our compounds would not have off target activities on human enzymes, however, to meet the comment of the Reviewer (and also question #10) we included the following sentences into the main text: "*... stereodynamics provides a greater adaptability for mutations for the inhibitors and thereby counter drug resistance induced by single-point mutations. It should, however, be mentioned that the greater adaptability of stereodynamic inhibitors might increase the risk of their binding to human metallo-enzymes and consequent unwished side effects. A higher risk of adverse off-target interactions is expected for compounds that sequester the zinc ion(s) of the metallo- β -lactamase binding site, such as EDTA and further chelators, whereas inhibitors specifically designed to bind zinc without chelation, such as taniborbactam, generally avoid human metalloenzymes.*"

2 – Microbiological assays

#4. Figure 2c. The authors should report the actual diameters of the growth inhibition zones measured with and without the inhibitor. Indeed, mentioning a percentage of variation of this value does not allow to asses the initial level of resistance of the tested strain, nor the actual synergistic activity of the inhibitor.

Response: We agree that relative increases in inhibition zone diameter do not allow conclusions to be drawn about the baseline level of resistance. However, given that the chosen assay is not intended to quantify resistance, but rather to demonstrate general inhibitory activity in the context of live bacteria, we still believe that the chosen graph format is more informative compared to inhibition zone diameters. For convenience, we provide the raw data as a supplementary figure (Figure S159).

Figure S159. Ampicillin-clavulanic acid zone diameters. AMC zone inhibition diameters were measured for *E. coli* TOP10 (wt, white column), *E. coli* TOP10xpCR4-Vim-2 (VIM-2, black column). Inhibitors 5a-l were added to AMC containing discs and inhibition zone diameters were recorded for *E. coli* TOP10 x pCR4-VIM-2 (red columns). Columns represent the mean of two independent experiments.

#5. Lines 397-399. The quantity of compound added to the discs should be mentioned here.

Response: Immediately after placing the disc onto the inoculated agar plate 10 µg compound were added per disc. This information has been added to the experimental section.

#6. Lines 400-408. It is unclear why the authors cloned a beta-lactamase gene in a beta-lactamase encoding plasmid, considering the variety of commercially available plasmids with other resistance markers. Indeed, the authors have to use sulbactam or tazobactam containing discs to assess the contribution of VIM-2 production on the susceptibility phenotype of the strain. Please comment.

Response: The plasmid was chosen since we knew that the backbone allows for high level expression of beta-lactamases in *E. coli* TOP10 (PMID: 38969718). In fact, unpublished data from our group indicates that for unknown reasons, the plasmid backbone impacts the expression levels even if beta-lactamases transcription is driven by cognate promoters. For that reason, we opted for pCR4 carrying a TEM beta-lactamase, and opted for using the highly efficient TEM inhibitor clavulanic acid.

#7. Lines 168-177. The title of this section is a strong assertion based on the available data. Indeed, only one recombinant (not a clinical isolate) strain was tested and no detailed susceptibility data are reported (see above). These data could be encouraging but should be put into context. Line 173, "in vivo" is not appropriate here. The authors mention 10 to 60% increase of the growth inhibition zone diameter, but Figure 2c does not show anything above roughly 25%. Please clarify. It should also be interesting to discuss the correlation between inhibitory potency and synergistic activity, especially considering the availability of inhibition data on OMVs. Indeed, compound 5e shows the highest potentiation while being a rather poor inhibitor.

Response: We thank the reviewer for lifting this point. To avoid confusion, we removed the term *in vivo*. The corrected sentence reads as "*This indicates that the compounds inhibit VIM-2 in bacteria.*" We have also adjusted the title of this session to "*Phosphonate-type metallo- β -lactamase inhibitors influence bacteria*" to avoid an overstatement.

We corrected the sentence about 10-60% increase of the growth inhibition zone diameter to 10-30% in the main text ("*The addition of inhibitors resulted in an increase in inhibition zone diameters, ranging from 10 to 30 % as compared to the control wild-type E. coli TOP10 without inhibitor (Figure 2c).*")

Regarding the correlation between inhibitory potency and synergistic activity, we added the following clarification to the discussion: "*The lack of correlation between the inhibitory activities detected on isolated enzymes, the permeabilities detected on outer membrane vesicles, and activities on bacterial growth (Figure 2) is not unexpected. The enzyme inhibition assay quantifies enzyme inhibition without further influencing factors, whereas membrane permeability measurements primarily indicate membrane permeability and do not quantify enzyme inhibition per se, whereas bacterial growth inhibition is influenced by a number of further factors, such as additional bacterial resistance mechanisms.*"

#8. Lines 156-167. Again, the conclusion seems to be strong based on the provided data. When compared to captopril, the level of phosphonate-mediated MBL inhibition in OMVs appears much lower. It is difficult to understand whether this is due to poor penetration or lower inhibition potency overall, because the IC₅₀ (or better the K_d) of captopril is not provided. It would be interesting to compare the level of inhibition in OMVs with the level of inhibition expected in similar conditions but with the free enzyme (again easily computed if the K_d is known). This would give a really interesting insight about the capability of the compounds to cross the outer membrane.

Response: We added the IC₅₀ (4.4 μ M) of L-captopril to help the reader.

To make a comparison to the freely accessible enzyme, we have recorded the enzyme activity data for OMVs permeabilized with Triton X-100 and have added the following text to the Supplementary Information:

*"In order to evaluate whether the difference in enzyme inhibition of compounds **5a-m** and L-captopril in the outer membrane vesicles, as shown in Figure 2b, originate from membrane penetration or inhibition potency, we also recorded the enzyme activity for OMVs permeabilized with Triton X-100. Whereas diffusion of both the nitrocefin as well as of the inhibitors 5a-m is limited by the membrane as they compete for the same pores, in the presence of Triton X-100 the vesicle membrane is solubilized and the metallo- β -lactamase becomes freely accessible (Figure S160). Since in presence of Triton X-100 the relative effects of the inhibitors are almost identical to the data with intact OMVs, this suggests that in first approximation compounds 5a-m all penetrate the vesicles equally well."*

Figure S160. Normalized relative activity of metallo- β -lactamases encapsulated in outer membrane vesicles in absence (buffer) and presence of inhibitors (a) in situ, and (b) following permeabilization of the membrane using Triton X-100. Error bars represent the standard deviation of four measurements. All compounds were used at a final concentration of 100 μ M. Relative activities based on the slope of the measured data were calculated as described in methods section."

We also added a reference to this news section of the Supplementary Information in the main text: "Upon permeabilization of the membrane with Triton X, the compounds showed comparable inhibitory activity on the free enzyme suggesting that the variation shown in Figure 2b is not due to different permeability of **5a-m** (Figure S160, Supplementary Information)."

3 – Discussion and implications

#9. The structural data are very convincing but are typically obtained in the presence of rather high concentrations of inhibitors. The authors interestingly observe the two enantiomers bound in the enzyme active site, but do not really discuss whether these two species could have similar inhibitory potencies or not (see e.g. lines 289-292).

Response: Due to their stereodynamic nature, these inhibitors cannot be separated (See Section 5, Supplementary Information and corresponding text in the manuscript ("*The final products are indeed stereodynamic, thus mixtures of inseparable interconverting enantiomers.*")) and their inhibitory activity cannot be determined separately. However, the comparable computational docking scores of the enantiomers to the VIM-2, NDM-1 and GIM-1 binding sites (Table S18

, Supplementary Information) along with the fact that crystals of the complexes of both enantiomers were obtained with VIM-2 when mixing a 1:1 mixture of the enantiomers with the protein suggest that both enantiomers bind with comparable affinity (otherwise only crystals with the stronger binding isomer would have been obtained).

To meet the Reviewers helpful request, we added the following text to the manuscript: "*Whereas the binding affinities of the inseparable enantiomers of 5a-m cannot be measured independently, the observation that VIM-2 complexes of both enantiomers of 5c-d were obtained when the protein was crystallized from a 1:1 mixture of the enantiomers suggests comparable affinity. This observation is corroborated by the computational docking which suggest binding of both enantiomers of 5a-m to the VIM-2, NDM-1 and GIM-1 binding sites with a ~180° rotated orientation yet with comparable binding scores, and comparable number of weak interactions to the protein binding site. The enantiomers may thus have comparable affinity and inhibitory affinity.*"

#10. The approach is very interesting, but the authors could seem a bit overenthusiastic regarding the fact that the stereodynamic compounds would mitigate resistance, even with a single aminoacid substitution. Indeed, in the complexes showing both enantiomers, the interactions appear to apparently involve similar amino acids (Tyr67, Trp87 seem important in both complexes). Please discuss. It would have been very interesting to measure the inhibitory activity of one compound with additional NDM- and VIM-type variants. Such data, although not equivalent to the characterization of laboratory-selected resistant mutants, could have been supportive of such statement. Lines 228-231. Could the relevant dynamic adaptability of the compounds, presented as an advantage here, be a limitation for the selectivity of the compounds (especially regarding human metallo-enzymes)? Please discuss.

Response: We have performed an extensive computational mutation study that confirmed the hypothesis the point mutations in the binding cleft primarily affect the binding of one or the other stereoisomer, but rarely that of both. This suggests the stereodynamics is a plausible tool to counteract resistance development due to single point mutations. See

our detailed response on question #23 below. Whereas experimental studies on VIM-2 and NDM-1 variants were not achievable for us for the current manuscript, we believe the computational insights provide sufficient evidence to support the presented hypothesis.

We agree with the Reviewer that the adaptability of the inhibitors may open for lower selectivity and possible binding to human enzymes. In order to meet this point, we added a comment to the main text as follows: *"It should, however, be mentioned that the greater adaptability of stereodynamic inhibitors might increase the risk of their binding to human metallo-enzymes and consequent unwished side effects. A higher risk of adverse off-target interactions is expected for compounds that sequester the zinc ion(s) of the metallo- β -lactamase binding site, such as EDTA and further chelators, whereas inhibitors specifically designed to bind zinc without chelation, such as taniborbactam, generally avoid human metalloenzymes."*

Minor comments

#11. Lines 44-46. This sentence is unclear. What are the "limited therapeutic options" intended by the authors? Polymyxins?

Response: We have examples and the corresponding references to help the reader: "This leaves very limited therapeutic options (including polymyxins, tigecycline and aminoglycosides) to treat β -lactam-resistant bacterial infections, which are suboptimal as they often confer serious toxic side effects including the constriction of bronchioles, anaphylaxis, and kidney and neurological disorders.^{9,13,14}"

#12. Lines 50-51. Both xeruborbactam and taniborbactam are in clinical development (phase 1 and phase 3, respectively). Reference 11 should be added,

Response: Corrected.

#13. Lines 56-57. This is well true but not limited to taniborbactam. Almost all antibiotics had resistance strains/mechanisms identified prior to their approval, starting with penicillin.

Response: We have rephrased the sentence as follows: *"This has been shown also for taniborbactam,¹⁹ against which resistance development has already been reported even before it could have reached the market."*

#14. Lines 77-78. It would be useful to remind the reader what are the structural peculiarities of GIM-1 that makes it so different from other subclass B1 enzymes.

Response: Following the Reviewer's suggestion, we added this information to the main text as follows: *"... but also against GIM-1 that possesses the most different active site*

compared to that of all other enzymes of the B1 metallo- β -lactamase family.^{32,33} It has a more constrained and narrower binding pocket as compared to other metallo- β -lactamases, and its binding site is composed of aromatic residues instead of hydrophilic ones, yet also containing a Ser119 and a Glu121 (BBL numbering³⁴, Table S33, Supplementary Information), which amino acid residues are not present at these positions in other metallo- β -lactamase enzymes."

#15. Line 171. Please specify the recipient strain (TOP10?).

Response: We added the following information to the experimental section: "*E. coli* TOP10 is a common *E. coli* lab strain allowing for high transformation efficiency (genotype F-mcrA D (mrr-hsdRMS-mcrBC) F80lacZDM15; DlacX74recA1 deoR araD139 D(ara-leu)7697 galU galK rpsL(StrR) endA1 nupG)."

#16. Lines 183-184. Very nice and convincing data and interpretation. Kinetic studies (e.g. Dixon plot) to confirm the competitive nature of the inhibition mechanism with a single compound could be an asset here.

Response: Following the suggestion of the referee, we performed Lineweaver-Burk analysis for **5g** on VIM-2, which confirmed competitive inhibition. This data is added to the supplementary material and the conclusion was added to the main text: "*A Lineweaver-Burk analysis⁴¹ confirmed competitive inhibition (Figure S161, Supplementary Information).*"

#17. Lines 205. The numbering used in the paper is confusing. Please use the consensus numbering scheme established for class B beta-lactamases throughout or at least provide the equivalence.

To make it easier for the reader, independently of background, we provide an additional table (Table S33, Supplementary Information) with conversion between the two common numbering systems, to help direct comparisons to previous data independent of the numbering scheme. In addition, we provide numbering corresponding to both systems in the main text.

#18. Figure 5. As such, not really useful (too small, absence of labelling).

Response: Figure 5 is corrected in the Manuscript.

#19. Line 364. EC50 or CC50 instead of IC50?

Response: We agree with the reviewer that the use of CC50 values is more appropriate. This is particularly important when pathogenic effects are measured, as in such cases the

IC50 refers to the inhibitory effect of the compound on the pathogen (therapeutic effect) in vitro. The test compound shows a dual effect: its inhibitory effect on the pathogen and its toxicity on the model cells used in the study. Therefore, we chose to follow the reviewer's suggestion and adjusted to report the toxicity values as CC₅₀ (see corrected Table S5, and adjusted text under Methods), which is defined as "cytotoxic concentration of the extracts required to cause death to 50% of viable cells in the host".

#20. Line 390. Please provide the molar concentration of the substrate.

Response: Adjusted.

Response to the comments of Referee-2:

The authors propose a new class of inhibitors to combat antimicrobial resistance, known as dynamically chiral phosphonic acids. As described in the manuscript, these compounds are easy to synthesize, can effectively penetrate bacterial membranes, and inhibit key MBL enzymes such as NDM-1, VIM-2, and GIM-1, without showing toxicity to human cells. In particular, these inhibitors are designed to target the catalytic zinc ions in MBLs. The authors highlight the adaptability of dynamically chiral phosphonic acids, as both stereoisomers can bind to MBLs, potentially enhancing their efficacy in preventing the development of antibiotic resistance in bacteria.

The research conducted by the authors demonstrates a well-structured approach, but not all of the analyses are comprehensive and could be improved. In conclusion, the results obtained are of interest to the scientific community. However, to further enhance understanding, certain points warrant clarification, which are outlined below.

Comments

#21. 1. On line 199, the authors state that the stereochemistry of the bound inhibitor depends on Asp118 or Asn210. Previous work highlights the relevance of Asn, which could be included: (1) de Seny et al., *Biochem. J.* 2002, 363, 687-696; (2) Garrity et al., *J. Biol. Chem.* 2004, 279, 920-927; (3) Medina et al., *ACS Catal.* 2022, 12, 36-47.

Response: We added the references as recommended, and have adjusted the text. (*"The importance of Asn residue on MBL inhibitor binding was previously also highlighted for homologous enzymes.⁵¹⁻⁵³ The number of hydrogen bonds to Asp118 or Asn210 (Asp120 and Asn233, BBL³⁴) differs between the (R)- and (S)- enantiomers of an inhibitor. For inhibitor **5d**, a hydrogen bond is formed in the (S)-enantiomer between the phenyl group and the backbone amide of Asp118 (3.3 Å, Asp120, BBL³⁴). In the (R)-enantiomer binding mode in subunit B, the phenyl- and the benzothiophene-moieties have switched places, such that the inhibitor is rotated by ~180°. Thus, the carboxamido moiety is forming the hydrogen bond with the backbone amide of Asp118 (3.1 Å, Asp120, BBL³⁴). The phenyl moiety forms an*

additional hydrogen bond with the side chain of Asn210 (3.1 Å, Asn233, BBL³⁴). The (R)-enantiomer differs from the (S)-enantiomer in the additional hydrogen bond of the carboxamido moiety. However, the predominant interaction between VIM-2 and the inhibitors' variable hydrophobic moiety takes place via non-bonded contacts. Between the phenyl ring or the thiophene/benzothiophene moieties of the inhibitors, π - π stacking may be observed with the aromatic ring of Phe62, Tyr67 or Trp87 (Phe61, Tyr67 and Trp87, BBL³⁴). Alternatively, a cation- π interaction of the benzothiophene moiety of the inhibitors **5c** and **5d** with Arg205 (Arg228, BBL³⁴) of VIM-2 is found. Arginine is known to have a tendency to form strong cation- π interactions.^{54,55} The guanidinium moieties of Arg205 (Arg228, BBL³⁴) and the benzothiophene moieties of **5c** and **5d** are parallel to each other with a shortest distance of about 3.5-3.7 Å in the crystal structures.”)

#22. 2. Line 205 indicates that a cation – π interaction is formed between the thiophene portion of the inhibitor and Arg205. The question is how this interaction can be probed, and an explanation could be included.

Response: The crystallographic structures of VIM-2 in complex with the inhibitors suggest a cation - π interaction between the Arg205 (228 BBL numbering) guanidinium moiety and the benzothiophene moieties of inhibitors **5c** and **5d**, which is based on the parallel arrangement of these and the corresponding distance of 3.5-3.7 Å (see Figure 4 main text). Cation - π interactions are very common in protein structures. According to a previous study, approximately 50% of arginine residues are in contact with multiple aromatic residues, and hence the observation of Arg205 (228 BBL numbering) being involved into a cation – π interaction is unsurprising.

This could certainly be proven by mutational analyses in combination with determination of the IC₅₀ of the mutant, however, considering the solid experimental evidence and this type of interaction being well-known to be commonly taking place in proteins, such a detailed investigation is out of the scope of this manuscript. To meet the recommendation of the Reviewer, we modified the discussion to highlight both the X-ray evidence and the previous literature: “...a cation- π interaction of the benzothiophene moiety of the inhibitors **5c** and **5d** with Arg205 (Arg228, BBL³⁴) of VIM-2 is found. Arginine is known to have a tendency to form strong cation- π interactions.^{54,55} The guanidinium moieties of Arg205 (Arg228, BBL³⁴) and the benzothiophene moieties of **5c** and **5d** are parallel to each other with a shortest distance of about 3.5-3.7 Å in the crystal structures.”

#23. 3. Line 230, the authors might consider including potential mutations, given the extensive theoretical and experimental data available on MBLs.

Response: Upon the recommendation of the Reviewer, we have performed a computational study on the influence of single point mutations on the binding affinity of the interconverting stereoisomers of our inhibitors. The results have been added to the

main text as follows: *“Stereodynamic inhibitors are predicted to counteract resistance upon point mutations. The fact that the interconverting stereoisomers of **5a-m** bind metallo- β -lactamases in different binding poses may counteract drug resistance induced by single-point mutations. In order to prove this, we conducted computational saturation mutagenesis screening on the amino acids located in the binding pocket of VIM-2 followed by molecular docking of both stereoisomers of inhibitors **5c** and **5d** to each mutant. The inhibitor binding affinities of 380 VIM-2 mutants were predicted using molecular docking, of which 37 reduced the binding affinity of the (R)-**5c** enantiomer, 105 of the (S)-**5c** enantiomer, and only 21 combinations affected the binding of both enantiomers (**Figure 6a**). For **5d**, 26 mutation combinations decreased the binding affinity of the (R)-enantiomer, 232 the binding of the (S)-enantiomer, and 19 combinations weakened the binding of both (**Figure 6b**). This observation suggests that the stereodynamics of the studied inhibitors decreases the risk for drug resistance development as specific single-point mutations of the target enzyme may weaken the binding of one of the enantiomers without affecting that of the other stereoisomer. Computational Saturation mutagenesis followed by molecular docking experiments for the inhibitors **5c**, **5d**, **5g** and **5j** to NDM-1 and GIM-1, and of **5g** and **5j** to VIM-2 (Figures S143-145, Supplementary Information) lead to the same conclusion as outlined for **5c** and **5d** when binding to VIM-2 mutants above. The two stereoisomers rapidly interconvert in solution, and accordingly the active stereoisomer will be selected and enriched upon binding to the mutated enzyme. Hence, stereodynamics provides a greater adaptability for mutations for the inhibitors and thereby counter drug resistance induced by single-point mutations.”* In addition, the results are shown in a visual form on Figures in the main text (Figure 6) and in the Supplementary Information (Figures S143-145)

#24. 4. The paragraph starting on line 244 discusses the results obtained from molecular dynamics (MD). The inclusion of harmonic restraints ($20 \text{ kcal mol}^{-1}\text{\AA}^{-2}$) throughout the entire classical MD simulation could lead to incorrect conclusions regarding ligand-protein interactions. Could the authors perform a molecular dynamics simulation with the entire system unrestrained to compare which interactions remain strongest during the simulation?

Response: We apologize for the misunderstanding caused by our unclear initial description of the MD calculation. The positional constraints ($20 \text{ kcal/mol/\AA}^2$) on the protein and ligand were applied only during the initial equilibration stages to ensure system stability and avoid any structural distortion. In the production phase of the molecular dynamics simulation all constraints were removed. As such, the production simulation results reflect the intrinsic interactions between the protein and ligand, free from external positional restraints. We have updated the description of the MD experiments in the Supplementary Information accordingly.

#25. 5. Line 255: The discussion regarding compound **5f** is confusing because it mentions the computational prediction of binding interactions, which aligns with the results

obtained from NMR. However, how did the authors calculate the binding interactions to reach that conclusion?

Response: To clarify how CSPs were determined, we added the sentence "*chemical shift perturbations were considered significant above the population mean plus at least one standard deviation^{49,50}*" to the manuscript. This is further discussed under section 11 in the Supplementary Information. Regarding the computations, we first used molecular docking to model the binding interactions of the VIM-2 complexes with **5c**, **5d**, **5g**, and **5j** (line 242-244). By aligning these docked structures with the available X-ray crystallographic data, we ensured that the docking protocol could accurately reproduce the experimentally observed binding modes. Next, we used the computational protocol to predict the binding modes for the other compounds and enzymes, for which no X-ray data is available. In terms of our predicted binding modes between **5f** and NDM-1, the interaction between zinc and the phosphate group is conserved. Also, we observed significant chemical shift perturbations of His122, Asp124, Gly188, Thr190, Ser191 from NMR, which are all residues close to the Zn coordination sites of the enzyme. Therefore, their chemical shift change during the titration may be attributed to the direct binding of **5f**. Other residues with significant chemical shift changes include Asp66, Met67 of loop 3, Val73, Ser75, Asn76 of β -sheet 3, Thr91, Trp93 of loop 5, and Lys211 of loop 10, which are either located around the pocket or in the same secondary structure, are either involved in direct interaction with **5f** or are indirectly affected. Hence, we draw a conclusion that the computational prediction of binding interactions aligns with the results obtained from NMR.

#26. 6. Line 263: It would be beneficial to include the work developed by Tripathi et al. in the list of references (Tripathi and Nair, ACS Catalysis 2015).

Response: We have added the proposed reference.

#27. 7. Paragraph starting on line 269: Each Zn is tetraordinated. Please correct the information indicated in parentheses. The three different loops are mentioned, but without a visual reference, it is difficult to understand how the residues are involved in inhibitor binding. The explanation of the loops should be improved, or a figure included to aid comprehension or complement Figure 5.

Response: We corrected the text regarding the coordination of Zn ions ("*Some of the amino acid residues responsible for Zn coordination (His118, His196, Asp120, His263, BBL)³⁴ and...*") to clarify that this sentence aims to indicate that some of the amino acids that were experimentally and computationally observed to be influenced by inhibitor binding coordinate to the Zn ions.

We improved Figure 5 adding the loop labelling and indicating the residues that interact with the inhibitors with a blue color.

General comments

#28. - Line 116 refers to Scheme 1, but it should refer to Figure 1b.

Response: Corrected.

#29. - Line 205, says tiophene instead of thiophene.

Response: Corrected.

#30. - The Supplementary Information is excessively large, and not all of the information is referenced in the manuscript. For example, on line 174, the computationally predicted ADME properties are mentioned as being in Section 16, but the correct section is Section 14. Additionally, the references to the SI file in the manuscript are not in consecutive order. Please review the references carefully, ensure they are correct, and remove any unnecessary information that is not relevant to the manuscript.

Reference: We have adjusted and corrected the Supplementary Information where needed.

Response to the comments of Referee-3:

This is a useful contribution, providing further data on phosphonates, which are well known metallo-enzyme inhibitors. The manuscript lacks comparisons and discussions with previous literature (supporting figures of comparisons needed, and discussion in the 'discussion' section). Some important details of inhibition are missing (see below). There is emphasis in the discussion on the importance of the stereochemistry, which is interesting but it is unclear how important this actually is to inhibition as the compounds are not purified as stereochemically pure, so some of the conclusions appear to be speculative and not fully demonstrated.

The work by Gulyas et al. describes the inhibition of three B1 MBLs by phosphonate inhibitors. Phosphonates are well known inhibitors of metallo-enzymes and metallo-beta-lactamases and the work presented here shows a new series of 13 phosphonates as μM inhibitors of three B1 MBLs. The structural data on their interactions with one, VIM-2, is interesting as it shows that the inhibitors can bind in a dual conformation, dependent on the stereochemistry. These data are worthy of publication, although there are some issues that should be addressed:

#31. 1. Importantly, there are not enough comparisons to previous work in this field. In particular, there should be comparisons against the structures described by Chen's group (<https://pubmed.ncbi.nlm.nih.gov/31483651/>), and Dmitrienko's group (<https://pubmed.ncbi.nlm.nih.gov/29485857/>), both in the text and in supporting figures. How does binding compare, which stereoisomers do they most closely resemble, what

inhibition do they show? Also how does binding compare to hydrolysed antibiotics (comparison figure against, for example, the recent VIM-2:biapenem work, PDB 6Y6J)?

Response: We extended the introduction with references to the above papers, and added a discussion of previous literature results -see our response on question #1 above.

Thereeto, we added a new subsection to the discussion comparing the binding mode of the presented inhibitors to those of previously published by others

"Comparison of the binding mode to those of previous inhibitors and of hydrolyzed antibiotics. The structure of a variety of heteroaryl phosphonates in complex with VIM-2 (PDB IDs 6D15, 6D16, 6D17, 6D18, 6D19, 6D1A, 6D1B, 6D1C, 6D1D, 6D1E, 6D1F, 6D1G, 6D1H, 6D1I, 6D1J, 6D1K, 6DD0, 6DD1, 6OR3, and 6NY7) are available in the Protein Data Bank (RCSB PDB), however, enzyme inhibitory activity (MIC, IC₅₀) has so far been disclosed for [(5,7-dibromo-2-oxo-1,2-dihydroquinolin-4-yl)methyl]phosphonic acid (PDB 6O5T) only. The X-ray crystallographic structure of this inhibitor indicates that the phosphonic acid of the latter binds directly to Zn2 (2.1 Å) and via a hydroxide bridge (2.0 Å) to Zn1 of the VIM-2 binding site (Figure S150c, Supplementary Information). At low pH, however, the phosphonate binds to both Zn ions directly. Further hydrogen bonds of this inhibitor with the main chain amide of Asn210 (3.5 Å, Asn233, BBL³⁴) and the side chain of Arg205 (3.5 Å, Arg228, BBL³⁴) are observed, but no further strong interactions to the binding site are seen. In contrast to the inhibitors presented in this study, the π -stacking interactions are not as obvious from the crystallographic data of 6O5T. In only a few of the literature compounds have at most one additional hydrogen bond with the enzyme. Overall, the phosphonic acid moiety of the inhibitor in the 6O5T structure interacts with the Zn ions of the enzyme in a similar way as the phosphonic acid of 5c, 5d, 5j, and 5g whereas the stereodynamics of these inhibitors result in different binding modes as compared to the that of 5,7-dibromo-2-oxo-1,2-dihydroquinolin-4-yl)methyl]phosphonic acid (Figure S150, Supplementary Information).

*6-(Phosphonomethyl)pyridine-2-carboxylic acid is a known inhibitor of the B1 metallo- β -lactamase IMP-1,⁴³ with the co-crystal structure PDB 5HH4 indicating that its phosphonic acid interacts with the Zn1 ion of the binding site via a hydroxide ion (2.6 Å) whereas its pyridine nitrogen (2.7 Å) and carboxylate group (2.3 Å) bind the Zn2 ion (**Figure S150d, Supplementary Information**) . The binding mode of 6-(phosphonomethyl)pyridine-2-carboxylic acid to IMP-1 lacks direct phosphonic acid – Zn contact and is thus fundamentally different from the binding poses of **5c**, **5d**, **5j**, and **5g** with VIM-2.*

*In the complex of VIM-2 with the hydrolysis product of biapenem (PDB 6Y6J), a carbapenem, the pyrrol (2.1 Å) and the carboxylate moiety (2.2 Å) of the hydrolyzed inhibitor closely interact with Zn2 of the binding site, whereas the carboxylate of the opened β -lactam ring interacts with Zn1 (1.9 Å) (**Figure S151, Supplementary Information**). Similar to the benzothiophene of **5c**, **5d**, **5j**, and **5g**, the heteroaromatic pyrazolotriazol moiety may engage in cation- π interaction. Further hydrogen bonds to the main chain amide of Asp118*

(3.1 Å, Asp120, BBL³⁴) and the sidechain of Asn210 (2.6 Å, Asn233, BBL³⁴) resembles the binding pose of **5c**, **5d**, **5j**, and **5g**."

#32. 2. Related to this, there should also be discussion and a comparison of the binding of these phosphonates with taniborbactam to VIM-2. This is important because the authors state that binding mimics the transition state of β -lactam hydrolysis, which is how boronates, like taniborbactam, bind. The relationship to the transition state is interesting but is not demonstrated directly. The structural reasoning with the TS should be explained further and justified.

Response: Upon the advice of the Referee, we added the following text to the results of the main text, along with supporting figures included into the Supplementary Material: *"The boron atom of boronate-type inhibitors, such as taniborbactam, react with the active site hydroxide ion and adopts an sp^3 hybridization state. This boron hydroxyl group interacts with the Asp118 and Asn210 (Asp120 and Asn233, BBL³⁴) amino acid side chains as well as with Zn1 of the binding site. Zn2, on the other hand, interacts with the carboxylate and the cyclic oxaborinane oxygen of taniborbactam. Accordingly, boronate-inhibitors mimic the tetrahedral intermediate of β -lactamase in the binding site of class B metallo- β -lactamases. Akin to the boronate-inhibitors, **5a-m** provide a tetrahedral phosphorus atom. The cyclic tetrahedral moiety of boronate inhibitors is conformationally restricted, resulting in similar orientations in both subunits for taniborbactam (**Figure S152c-d, Supplementary Information**), whereas inhibitors **5c**, **5d**, **5j**, and **5g** are less restricted and can adapt to the active site by different orientations (**Figure S152a-b, Supplementary Information**). The tetrahedral transition geometry is a transient state in the hydrolysis of β -lactam antibiotics. Inhibitors that stably adopt this geometry may therefore prevent the progress of hydrolysis and inhibit the enzyme."*

#33. 3. Some of the discussion of the different modes of binding for the different stereoisomers is not fully justified. In particular, it is unknown if this is important for binding different MBLs (there is no data to show the inhibition of each stereoisomer, because they are purified as a mixture). Also, there is no evidence that these could "counteract resistance development", as only 3 enzymes are tested.

Response: Due to the stereodynamic nature of these inhibitors, their stereoisomers rapidly interconvert and are impossible to separate (see Section 5 "Chiral HPLC" in Supplementary Information describing our attempts to separate them on 12 chiral columns with 2 solvent combinations on normal phase HPLC, with 7 chiral columns with 4 solvent combinations on reverse-phase HPLC, with 11 chiral columns with 2 solvent combinations with basic SFC, and 6 chiral columns with 2 solvent combinations with acidic SFC) and study them separately. We therefore cannot generate experimental inhibitory data for the separate stereoisomer.

However, in order to meet the comment of the Reviewer, we performed an extensive computational mutation study followed by molecular docking of the two stereoisomers of

selected inhibitors to VIM-2, NDM-1 and GIM-1 and demonstrated that single-point mutations primarily affect the binding of only one of the stereoisomers but not of both. Please see detailed response at questions #23 above. We believe that a concept shown using 3 different enzymes is a good proof of concept indication for a hypothesis to be likely useful also for a larger set of enzymes within the same subclass of beta-lactamases.

#34. #4. Do the compounds exhibit any time-dependent inhibition? i.e. what happens to the IC₅₀s when the protein is incubated for longer? The authors also mention no zinc chelation is observed in the NMR (line 184), but it would be useful here to say over what timescale this is (minutes, hours?). It is possible that zinc chelation may be time-dependent (over hours and not minutes).

Response: We have studied the inhibition with NMR over a 10 hours time period, and have not seen signs of time-dependent inhibition. This has been added to the manuscript (*"The pattern and the magnitude of the chemical shift changes are incompatible with Zn(II) depletion of the enzyme (over the 10 hours experiment time)."*) The enzyme inhibition assay was run with an incubation time of 5 min before measurement, then measured for approximately 10 min to completion (substrate depletion).

#35. 5. For the structures of VIM-2 with 5j and 5g, the crystal table needs to show stats for the resolution range that was refined against (i.e. high resolution of 1.4 and 1.36, respectively).

Response: Corrected.

#36. #6. For the IC₅₀s the authors should explain how the standard error of the mean was calculated (it suggests at least 3 independent IC₅₀ values were calculated – not just technical replicates – which is not mentioned in the methods), and why this is presented rather than the standard error of the logIC₅₀, which is more accurate for non-linear regression.

Response: The SEM of the IC₅₀ values were calculated based on the SEM of logIC₅₀. We added the logIC₅₀ values and their standard errors that were used to calculate the reported error, to bring clarity (Table S1, Supplementary Information)

#37. 7. S144, S145, S146 are barely readable due to their size and low resolution, but are important for understanding the quality of the crystallographic data. These should be split up so that the images are viewable and at least the electron density can be seen properly (particularly at the resolution in the PDF).

Response: We have updated these figures in the Supplementary Information.

#38. Line 51 clarify that it is the boronate hydroxy group that binds.

Response: Corrected.

#39. 9. Include the ligand RSCCs (from PDB validation reports) in the text (including of both binding conformations), or as a supporting table.

Response: We show electron density for every ligand in the figures of crystallographic structures so readers can assess the fit of the ligand to the density. In addition, we now provide RSCC values in Table S25 with a value for every bound enantiomer to aid readers obtaining a better overview of the electron density fit.

Response to the comments of Referee-4:

In this manuscript, the authors present the synthesis and biological activity of chiral α -amino phosphonic acid as metallo-beta lactamase inhibitors. The binding mode of selected inhibitor-enzyme complexes was evaluated by NMR and X-ray crystallography and molecular modeling. The inhibitors were active in enzymatic and bacterial assays with the best inhibitors showing activity in the low micromolar range, with no detectable cytotoxicity.

The main claim of the work is that the α -amino phosphonic acids designed by the authors can stereochemically adapt via racemisation to the binding pocket of different enzymes and as such provide potentially broader enzyme coverage. The concept of stereochemically adaptable inhibitors is intriguing and relevant for further inhibitor design and should be considered novel in the context of MBL inhibition and small-molecular inhibitors.

However, before publication, the authors should clarify the following:

#40. • All the inhibitors are prepared as racemic mixtures containing both enantiomers of the inhibitors. Can the authors be sure that the α -amino phosphonic acids are racemized i.e. are "dynamically chiral" under assay conditions? I cannot find documentation for the dynamic behavior in the manuscript. For comparison, amino acids in the acid form are not particularly prone to racemization under mild conditions (Bada, J.L., Kinetics of racemization of amino acids as a function of pH, *J. Am. Chem. Soc.* 94(4):1371–1373, 1972; Bada, J.L. and Schroeder, R.A., Amino acid racemization reactions and their geochemical implications, *Naturwissenschaften* 62:71–79, 1975).

Response: The α -CH of natural amino acids is less acidic as compared to a benzylic-CH adjacent to electron withdrawing phosphonic acid and amide moieties. The influence of the introduction of a phenyl group next to the CH- α of an amino acid has been shown by Smith and Sivakua in *J.Org. Chem.* 1983, 48, 627 (DOI: 10.1021/jo00153a001) to

amplify racemization rate by a factor of 40 000, corresponding to a 8 kcal/mol energy difference. A 28 kcal/mol activation barrier (Table 2 in the above paper), corresponding to a glycine, means years for racemization at room temperature whereas a 20 kcal/mol of phenyl-glycine leads to full racemization with minutes. We added this information to the main text as follows: *"Introduction of a phenyl group adjacent to the α -proton of an amino acid has previously been shown to decrease the activation barrier of racemization by ~8 kcal/mol, accelerating racemization by a factor of 40 000.⁴¹"*

Section 5 of the Supplementary Information provides experimental evidence for the stereoisomers of **5** not being possible to separate by HPLC, despite having tested a very extensive set of columns and conditions (please see also our response on question #33 above). The reader is directed to the details of the chromatographic work by the sentence *"The final products are indeed stereodynamic, thus mixtures of inseparable interconverting enantiomers (see details on the chiral separation in the **Supplementary Information, Section 5**)."*

We further added the sentence *"We note that a drug candidate encompassing a mixture of stereoisomers that do not interconvert yet bind to the active site of an enzyme with different binding modes may provide a comparable advantage."* to the results section.

#41. • Line 88/89: The authors write "Once bound to the enzyme's active site, the phosphonic acid remains strongly coordinated to the Zn ions....» does this imply that the inhibition is irreversible? Have the authors tested this by e.g. jump-dilution experiments?

Response: The binding is not covalent and hence the strong coordination does not mean irreversible inhibition. We have not claimed this in the original version, but for clarification we adjusted the text adding the following comment: *"The obtained NMR data indicates reversible inhibition of the enzyme."*

Minor comments:

#42. * The authors use the terms "dynamically chiral" "stereodynamic binding" and "stereodynamic inhibitor" in the manuscript. Even though, there is a reference (line 105) in the text, the manuscript would benefit from defining the concept briefly in the text.

Response: We added an explanation to the introduction, as suggested: *"Drug candidates possessing a dynamically chiral stereocenter rapidly interconvert between multiple chiral configurations under physiological conditions, which adaptation of their stereochemistry for binding to a target. Stereodynamic compounds may convert to the stronger binding stereoisomer to bind to an enzyme's active site, or may allow binding with different, stereochemistry-dependent binding modes to the same enzymatic site. We anticipate that stereodynamics may facilitate inhibitor candidates' binding to a variety of metallo- β -lactamases, potentially even to mutated variants".*

#43. * Line 288: "novel and innovative" should be rewritten.

Response: Corrected.

#44. The experimental section is carefully prepared and appears as reproducible.

Response: We are grateful for noting the effort to provide all relevant data in the SI.

Response to the comments of Referee-1:

The authors provided detailed answers to the referees' comments, some convincing, some a bit less (with also a few conceptual mistakes, such as in page 4, vaborbactam is not an MBL inhibitor). Disappointingly, many new and interesting data were included in the SI rather than in the manuscript. More importantly, the authors provided evidence to support the competitive inhibition of at least one compound (Fig. S161). Please note that in Table S35, what is reported as the K_m value (which does NOT vary) is actually the apparent K_m value (K_m app or K_m'). I still believe that the way the authors present their AST data is not appropriate (mentioning that the purpose of such experiments is "not to quantify resistance", one might then ask what "antimicrobial susceptibility testing" means). These data (Fig. 2C) are presented in an unconventional and misleading way, as the analysis of the raw data does not allow to strongly assert that "the compounds inhibit VIM-2 in bacteria", considering that very limited, if any, AMP/SUL potentiation is observed (in sharp contrast with "We agree that relative increases in inhibition zone diameter do not allow conclusions to be drawn about the baseline level of resistance" as stated in the rebuttal). The absence of significant synergistic activity is obviously not a problem for publication (as it is often the case for exploratory research) but data should be analyzed and conclusions drawn with objectivity. Also, "bacterial growth inhibition is influenced by a number of further factors, such as additional bacterial resistance mechanisms" might well be true but using a laboratory strain producing the MBL, one might wonder what these additional mechanisms could be. The addition of the K_i values (Table S2) show that the inhibitors are indeed not very potent, this might also be the reason underlying the lack of potentiation on *E. coli* cells. Overall, the inhibitor design and overall approach remains very interesting.

Response:

We apologize for the mistake in the previous response letter, in which we mistakenly mixed up xerubobactam with vaborbactam. As this mistake was in the response letter and not in the manuscript, no related changes have been made to the manuscript.

We fully agree with the referee and have changed K_m to K_m' in Table S6 of the Supplementary Information.

Upon the comment of the Reviewer, we have included an alternative representation of the zone inhibition assay to Figure 2 as subfigure 2d. This data complements the enzyme inhibition assay and the membrane permeability assay, shown in Figure 2a and 2b, respectively. The key observation from the zone inhibition assay is that upon adding compounds **5a-1**, a 10-30% change in the inhibition zone diameter is observed, compared to the control samples. This data should certainly not be over-interpreted. It corroborates the conclusions drawn from the enzymatic and permeability assays, and indicate that compounds

5a-I show some growth inhibition. These compounds are certainly not directly clinically applicable, which the manuscript does not state either.

We wish to underline that the key point of this manuscript is not to present clinically applicable MBL inhibitors, but rather to introduce the concept of 'dynamically chiral enzyme inhibition' as well as a group of phosphonic acid-type MBL inhibitors. To meet the comment of the Referee, we have modified the sentence "*This indicates that the compounds inhibit VIM-2 in bacteria.*" to "*This indicates that the compounds show some VIM-2 inhibition in bacteria, corroborating the outcome of the enzyme inhibition and membrane permeability assays. The extent of inhibition is however not directly clinically applicable.*" We have also updated Figure 2 in order to show the result of the zone inhibition assay in an alternative way.

Manuscript ID: COMMSCHEM-24-0467-T

Report on the paper “*Dynamically chiral phosphonic acid-type metallo- β -lactamase inhibitors*” by Máté Erdélyi et al.

The authors propose a new class of inhibitors to combat antimicrobial resistance, known as dynamically chiral phosphonic acids. As described in the manuscript, these compounds are easy to synthesize, can effectively penetrate bacterial membranes, and inhibit key MBL enzymes such as NDM-1, VIM-2, and GIM-1, without showing toxicity to human cells. In particular, these inhibitors are designed to target the catalytic zinc ions in MBLs. The authors highlight the adaptability of dynamically chiral phosphonic acids, as both stereoisomers can bind to MBLs, potentially enhancing their efficacy in preventing the development of antibiotic resistance in bacteria.

The research conducted by the authors demonstrates a well-structured approach, but not all of the analyses are comprehensive and could be improved. In conclusion, the results obtained are of interest to the scientific community. However, to further enhance understanding, certain points warrant clarification, which are outlined below.

Comments

1. On line 199, the authors state that the stereochemistry of the bound inhibitor depends on Asp118 or Asn210. Previous work highlights the relevance of Asn, which could be included: (1) de Seny et al., *Biochem. J.* 2002, 363, 687-696; (2) Garrity et al., *J. Biol. Chem.* 2004, 279, 920-927; (3) Medina et al., *ACS Catal.* 2022, 12, 36-47.
2. Line 205 indicates that a *cation* – π interaction is formed between the thiophene portion of the inhibitor and Arg205. The question is how this interaction can be probed, and an explanation could be included.
3. Line 230, the authors might consider including potential mutations, given the extensive theoretical and experimental data available on MBLs.
4. The paragraph starting on line 244 discusses the results obtained from molecular dynamics (MD). The inclusion of harmonic restraints ($20 \text{ kcal mol}^{-1} \text{ \AA}^{-2}$) throughout the entire classical MD simulation could lead to incorrect conclusions regarding ligand-protein interactions. Could the authors perform a molecular dynamics simulation with the entire system unrestrained to compare which interactions remain strongest during the simulation?

5. Line 255: The discussion regarding compound 5f is confusing because it mentions the computational prediction of binding interactions, which aligns with the results obtained from NMR. However, how did the authors calculate the binding interactions to reach that conclusion?
6. Line 263: It would be beneficial to include the work developed by Tripathi et al. in the list of references (Tripathi and Nair, ACS Catalysis 2015).
7. Paragraph starting on line 269: Each Zn is tetracoordinated. Please correct the information indicated in parentheses. The three different loops are mentioned, but without a visual reference, it is difficult to understand how the residues are involved in inhibitor binding. The explanation of the loops should be improved, or a figure included to aid comprehension or complement Figure 5.

General comments

- Line 116 refers to **Scheme 1**, but it should refer to **Figure 1b**.
- Line 205, says tiophene instead of thiophene.
- The Supplementary Information is excessively large, and not all of the information is referenced in the manuscript. For example, on line 174, the computationally predicted ADME properties are mentioned as being in Section 16, but the correct section is Section 14. Additionally, the references to the SI file in the manuscript are not in consecutive order. Please review the references carefully, ensure they are correct, and remove any unnecessary information that is not relevant to the manuscript.